# Grid-like entorhinal representation of an abstract value space during prospective decision making

Alexander Nitsch [1] ✉, Mona M. Garvert[1,2,3,4], Jacob L. S. Bellmund [1],
Nicolas W. Schuck [2,3,5] & Christian F. Doeller [1,6,7,8] ✉

How valuable a choice option is often changes over time, making the prediction of value changes an important challenge for decision making. Prior studies identified a cognitive map in the hippocampal-entorhinal system that encodes relationships between states and enables prediction of future states, but does not inherently convey value during prospective decision making. In this fMRI study, participants predicted changing values of choice options in a sequence, forming a trajectory through an abstract two-dimensional value space. During this task, the entorhinal cortex exhibited a grid-like representation with an orientation aligned to the axis through the value space most informative for choices. A network of brain regions, including ventromedial prefrontal cortex, tracked the prospective value difference between options. These findings suggest that the entorhinal grid system supports the prediction of future values by representing a cognitive map, which might be used to generate lower-dimensional value signals to guide prospective decision making.

Optimal decision making relies on predictions of future values associated with choice options. For example, if you were to invest in stocks, you would want to choose those stocks which are likely to be more valuable than others in the future. In particular, this implies that you should be able to predict if and when one stock becomes more valuable than another and choose accordingly to maximize long-term reward. Crucially, such prospective decision making requires an appropriate neural representation of the relation between changing and future values of choice options.

Prior studies established a role for parts of the ventromedial prefrontal cortex (vmPFC) and orbitofrontal cortex (OFC) as well as the ventral striatum in tracking the value difference between the chosen and the unchosen option during decision making[1–11]. Correct decisions

in many previously used tasks depended primarily on updating values based on experience[12]. However, many decisions, such as in the introductory example of the stock market, require recognizing trends and extrapolating values into the future. In such scenarios, dorsal anterior cingulate cortex (dACC) has been implicated in comparing recent and past reward rates, allowing for trend-guided choices based on expected future rewards[13,14].

Prediction of future values is enabled by an internal model, which represents transitions between states and reward contingencies in an environment or task. Reliance on an internal model has been referred to as model-based decision making, and can lead to distinct value computations found in the dorsomedial prefrontal cortex (dmPFC)[15–17]. Moreover, the hippocampus has been implicated in model-based and

[1]Max Planck Institute for Human Cognitive and Brain Sciences, Leipzig, Germany. [2]Max Planck Research Group NeuroCode, Max Planck Institute for Human Development, Berlin, Germany. [3]Max Planck UCL Centre for Computational Psychiatry and Aging Research, Berlin, Germany. [4]Faculty of Human Sciences, Julius-Maximilians-Universität Würzburg, Würzburg, Germany. [5]Institute of Psychology, Universität Hamburg, Hamburg, Germany. [6]Kavli Institute for Systems Neuroscience, Centre for Neural Computation, The Egil and Pauline Braathen and Fred Kavli Centre for Cortical Microcircuits, Jebsen Centre for Alzheimer's Disease, Norwegian University of Science and Technology, Trondheim, Norway. [7]Wilhelm Wundt Institute for Psychology, Leipzig University, Leipzig, Germany. [8]Department of Psychology, Technical University Dresden, Dresden, Germany. ✉e-mail: nitsch@cbs.mpg.de; doeller@cbs.mpg.de

value-based decision making[18–24]. Interestingly, Vikbladh et al.[22] found that the hippocampus serves as a common neural substrate for both model-based decision making and place memory in spatial navigation. A possible mechanism by which the hippocampus could support both model-based decision-making and spatial navigation is via the formation of cognitive maps.

Cognitive maps encode relationships between states in the world in a map-like format[25–32]. Neurally, cognitive maps are assumed to rely on the activity of spatially tuned cells in the hippocampal-entorhinal system. For example, during spatial navigation, place cells in the hippocampus exhibit increased firing at a particular location within an environment[33]. Grid cells in the adjacent entorhinal cortex fire at multiple locations within an environment and these locations form a hexagonal grid[34]. Together, these cells enable self-localization and geometric computations supporting spatial navigation, e.g., the computation of distances and directions[35,36]. Beyond spatial navigation, recent studies have shown hippocampal-entorhinal map-like and grid-like representations of more abstract information, e.g., in graph-like structures[37] as well as in feature and concept spaces[38–43]. Therefore, hippocampal-entorhinal cognitive maps have been suggested to provide a more general mechanism for organizing information, allowing for adaptive decision making[25,26,44–46]. For example, two recent studies showed distance- and grid-like representations for novel inferences during decision making in a two-dimensional map of social hierarchies[47,48].

In decision making, states in the world and values are usually considered different entities, i.e., values (rewards) are received after performing an action in a given state. However, it is conceivable that values constitute states themselves, which can be represented in a cognitive map. In line with this notion, Bongioanni et al.[49] demonstrated first evidence for a grid-like representation of an abstract value space defined by reward magnitude and probability in macaques. While choice options in previous studies[47–49] were static with regard to their locations in the abstract space, an interesting question is whether the same map-like representation would code for values of options changing over time. By facilitating computations of directions of and distances between value changes over time, such a cognitive map could enable efficient prediction of future values for prospective decision making. This map could then be used to read out resulting values and generate lower-dimensional signals of the value difference between options and their identities for choices. First evidence for hippocampal neurons encoding position in a value space spanned by changing reward probabilities has been demonstrated in macaques[50]. However, it remains elusive whether an entorhinal grid-like representation would encode changing values during prospective decision making in humans.

Here, we aimed to investigate whether the entorhinal cortex integrates relational information about changing values during prospective decision making using a grid-like representation of an abstract value space. To address this question, we combined functional magnetic resonance imaging (fMRI) with a prospective decision making task which required participants to integrate values in an abstract two-dimensional value space. Our behavioral results show that participants integrated and extrapolated changes along the two value dimensions to guide prospective choice, indicating they formed a map of the relationships between options. Crucially, while participants traversed the abstract value space along trajectories, the entorhinal cortex exhibited a grid-like representation, suggesting the formation of a cognitive map. A network of brain regions, including the ventromedial and dorsal prefrontal cortex, tracked not only the value difference between options during choices, but also particularly the prospective value component.

## Results

### Participants integrate and extrapolate value changes for prospective choices

We monitored whole-brain activity using fMRI while 46 participants performed a prospective decision making task (Fig. 1). The task required participants to maximize reward by tracking and predicting values (i.e., reward magnitudes) associated with two choice options. Each trial (Fig. 1a) consisted of an observation phase and an active choice. During the observation phase, participants viewed the two options along with their changing values over a sequence of time points. They were instructed to carefully track the value changes to be able to predict the options' values at the next time point. After 3−5 observed time points, participants were asked to choose the option with the higher value at the next time point (choice time point). Correct choices were translated into a monetary bonus for participants, which was based on the options' values.

More specifically, the two options were represented by the same four category-specific stimuli, which were mapped onto the two options (e.g., face/tool signaled option A, while hand/scene signaled option B). The stimulus mapping remained constant throughout the task and participants were informed about it before. Across time points, the two value-congruent stimuli of a given option alternated.

Crucially, a sequence of time points formed a trajectory through an underlying abstract two-dimensional value space, with the dimensions of the space corresponding to the values associated with the two options (Fig. 1b, Supplementary Fig. 1). In this space, the 45°-diagonal represented locations where the two options had the same values. Trajectories crossing the 45°-diagonal therefore involved a switch in which of the two options was more valuable. Tracking value changes over time, essentially recognizing the direction of and distances along a trajectory, allowed for prediction of future values and therefore detection of switches.

Participants' overall performance of the task, as indicated by choices of the more valuable option, was high (Fig. 2a, M = 87.70%, SD = 6.48%). If participants considered value changes over time for their choices, they should have detected switches of the more valuable option from one time point to the next. Indeed, they detected switches significantly more often than expected by chance (Fig. 2b; $t(45) = 10.82$, $p < 0.001$). Apart from the switch time point, the more valuable option was the same as at the preceding time point and participants could simply stay with that option. We therefore compared switch performance with the time points before (pre) and after (post) a switch. This comparison revealed a significant effect of time point (Fig. 2b; $F(2,90) = 35.93$, $p < 0.001$). Post-hoc pairwise tests indicated significantly reduced performance for both pre and switch compared to post (pre: $t(45) = -10.72$, $p < 0.001$; switch: $t(45) = -7.35$, $p < 0.001$) but no significant difference between pre and switch ($t(45) = -0.90$, $p = 0.40$; all with $\alpha = 0.016$, Bonferroni-corrected for three comparisons; controls for pre and post individually against chance: pre $t(45) = 11.02$, $p < 0.001$, post $t(45) = 45.17$, $p < 0.001$; $M \pm SD$: pre $73.91 \pm 14.71\%$, switch $77.09 \pm 16.98\%$, post $95.52 \pm 6.83\%$). Similarly, we observed a significant effect of time point on reaction times (Supplementary Fig. 2a; $F(2,90) = 60.65$, $p < 0.001$; post hoc pairwise tests with $\alpha = 0.016$: pre-post: $t(45) = 9.85$, $p < 0.001$, switch-post: $t(45) = 1.74$, $p = 0.09$, pre-switch: $t(45) = 8.33$, $p < 0.001$). This pattern of results suggests that participants successfully detected switches of the more valuable option and may even have over-extrapolated the value changes, leading to earlier switches than optimal.

As switches are induced by the 45°-diagonal of the value space, we tested more continuously how performance is influenced by the distance between the choice location and the diagonal using participant-specific logistic regressions. We found that the likelihood of correct choices increased with increasing distance of the choice location from the diagonal (Fig. 2c, d; $t(44) = 8.03$, $p < 0.001$; $M \pm SD$ $1.64 \pm 1.37$ arb. units). As a control, we tested the same relationship using only choices in switch trajectories where locations lay inherently closer to the diagonal (Fig. 2c; $t(44) = 6.60$, $p < 0.001$; $M \pm SD$ $0.81 \pm 0.83$ arb. units). This suggests that the closer choice locations were to the 45°-diagonal and hence the more similar the options' values became, the more difficult the choices became for participants.

Next, we investigated whether a reinforcement learning model which captured the prospective nature of the task, i.e., the value changes over time, fitted participants' choice behavior better than a model that did not. To this end, we modified a Rescorla–Wagner model[12] so that it updated value estimates within a trial based on prediction errors and additionally value changes over time points:

$$V_{TP+1} = V_{TP} + \alpha^* (O_{TP} + C_{TP} - V_{TP}) \text{ with } C_{TP} = O_{TP} - O_{TP-1}, \quad (1)$$

whereby $V_{TP}$ and $V_{TP+1}$ are values at the current and next time points, respectively, $O_{TP}$ is the outcome at the current time point, $C_{TP}$ reflects how the value has changed from the previous to the current time point and $\alpha$ is the learning rate (free parameter of the model). In essence, this prospective Rescorla–Wagner model does not only update the expected value to the outcome just observed, but learns which outcome to expect given the past history of changes. We compared this to the original Rescorla–Wagner model which does not consider value changes over time points:

$$V_{TP+1} = V_{TP} + \alpha^*(O_{TP} - V_{TP}) \quad (2)$$

As expected, the prospective Rescorla-Wagner model fitted the data better than the original Rescorla-Wagner model (Fig. 2e; model comparison per AIC: $t(45) = -8.71$, $p < 0.001$, with $\alpha = 0.01$, Bonferroni-corrected for five tests including alternative models; $M \pm SD$: prospective $51.81 \pm 17.68$, original $63.63 \pm 11.51$). Initially, we constrained the learning rate of the prospective model to the range between 0 and 1, with 1 reflecting full updating according to prediction errors and value changes. We observed a ceiling effect for the learning rate, with many participants having learning rates of 1 ($M = 0.94$, $SD = 0.12$). For

this reason, we removed the upper bound of the learning rate and observed learning rates slightly above 1 on average, suggesting slight over-updating in line with the performance reduction at the pre time point described above (Supplementary Fig. 2g; $M = 1.09$, $SD = 0.25$; model comparison of unbound and bound model per AIC: $t(45) = -3.96$, $p < 0.001$). The learning rate correlated positively with performance at the switch time point (Supplementary Fig. 2h; $r(44) = 0.55$, $p < 0.001$) but negatively with performance at the pre time point (Supplementary Fig. 2i; $r(44) = -0.44$, $p = 0.003$), reflecting the advantage and disadvantage of over-updating. In addition, we implemented a set of alternatives for the prospective model, e.g., with a separate learning rate for the change term or with a term for an expected prediction error (see Methods). The prospective model described above fitted the data better than all alternatives (Supplementary Fig. 2j; all $p < 0.001$). These modeling results confirm and extend our previous pre-switch-post performance analysis by showing that participants indeed extrapolated value changes for prospective choices, though to a slightly larger extent than optimal, presumably causing too early switches.

Lastly, we reasoned that if prospective decision making is supported by a cognitive map similar to spatial navigation, then participants with better navigational abilities may also perform better in our prospective decision making task. To investigate whether this is the case, we tested whether behavior in our task correlated with participants' self-reported navigational abilities and preferences (as measured by the Santa Barbara Sense of Direction Scale (SBSOD) questionnaire[51], completed in the last part of the study). We observed a significant positive correlation between the learning rate of the prospective Rescorla–Wagner model and self-reported navigational abilities and preferences (Fig. 2f; correlation with learning rate: $r(44) = 0.34$, $p = 0.02$; correlation with overall performance:

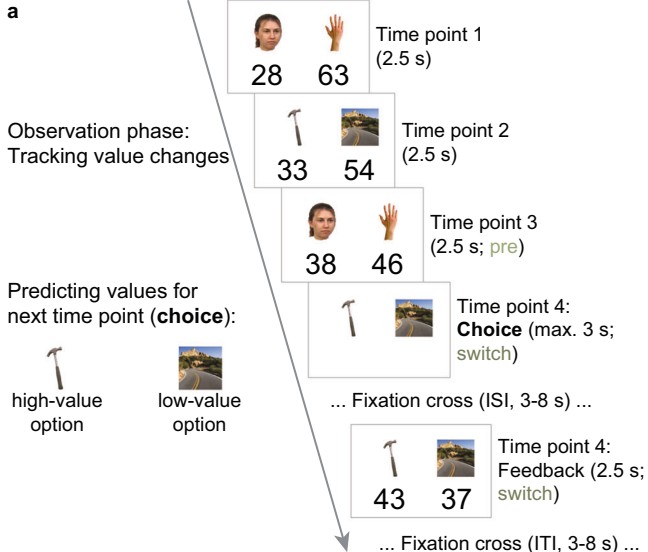

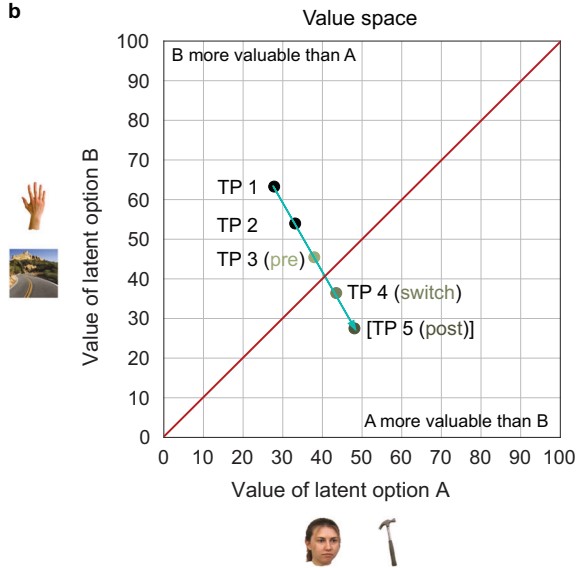

**Fig. 1 | Design of prospective decision making task.** Participants were instructed to track and predict changing values of two latent options A and B. Each latent option was signaled by one of two associated images, e.g., a face or a tool for option A and a hand or a scene for option B. Participants were instructed which images signaled the same option before the task. **a** Example trial, consisting of an observation phase and an active choice. During the observation phase, participants viewed the two options along with their changing values over a sequence of 3 time points (TPs). Across TPs, the two value-congruent images of a given option alternated. Subsequently, participants were asked to choose the option with the higher value at the next TP (choice TP). For their choice, participants had to consider how the values changed over time and accordingly how they will have changed towards the choice TP. Finally, they received feedback about the actual values at the choice TP. Note that in this example trial, the more valuable option changed from hand / scene (option B) in the beginning of the trial to face / tool (option A) at the choice

TP. To facilitate fast tracking of the value changes across TPs, options were displayed on the same side of the screen during the observation phase but sides were random during choices. **b** A trial with its sequence of TPs formed a trajectory through an underlying abstract two-dimensional value space, with the dimensions corresponding to the values associated with the two options. Each TP corresponded to a particular location in the value space, depicted by dots. The arrow in turquoise depicts the trajectory through these locations. Trajectories crossing the 45°-diagonal of the space (red) involved a switch of the more valuable option. Choices sampled different TPs across trajectories (trials). The first TP after the diagonal is referred to as the switch and the TPs before and after the switch as pre and post. Note that the trial ended after the choice TP (see **a**) and the post TP is added to the trajectory for illustration. Stimuli taken from publicly available stimulus datasets (see Methods)[77–82].

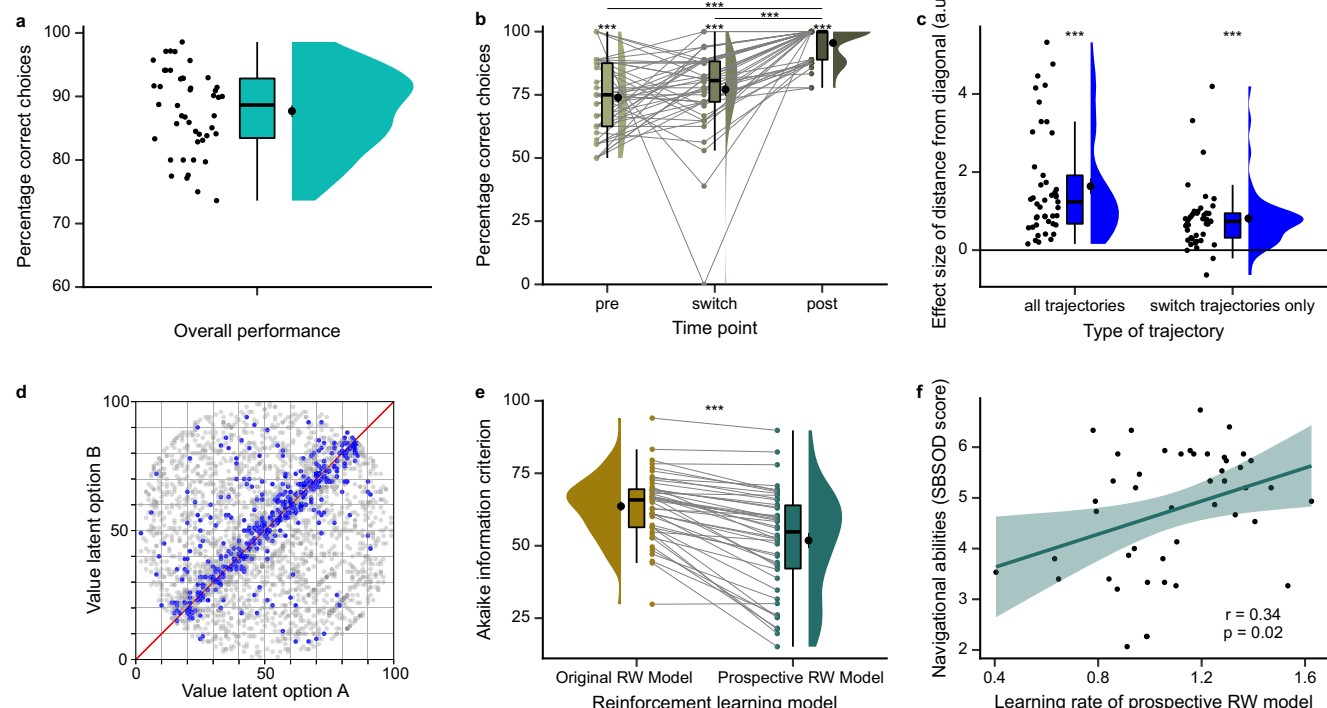

**Fig. 2 | Participants integrate and extrapolate value changes for prospective choices. a** Overall performance across all trajectories of the task. **b** Performance in switch trajectories, at the time point before the switch (pre), of the switch (switch) and after the switch (post). Performance at all time points was significantly above chance (one-sample $t$-tests against chance (50%), all $p < 0.001$), suggesting that participants succeeded in detecting switches. Performance at pre and switch was significantly lower than at post time points (repeated measures ANOVA with Bonferroni-corrected post-hoc related-samples $t$-tests, all $p < 0.001$ except for comparison pre-switch with $p = 0.40$). **c** Effect of the distance between the choice location and the 45°-diagonal of the value space on performance, separately for all trajectories (left) and only switch trajectories as a control (right). In both cases, effect sizes estimated by participant-specific logistic regressions are significantly positive (one-sample $t$-tests, all $p < 0.001$), indicating that the likelihood of correct choices increased with increasing distance from the diagonal. **d** Visualization of the distance-from-the-diagonal effect in **c**. Blue dots depict incorrect choice locations across participants, clustering around the 45°-diagonal. Gray dots depict correct

choice locations. **e** Reinforcement learning model comparison. The Akaike information criterion (AIC) is significantly lower (better model fit) for the prospective Rescorla–Wagner model (right) compared to the original Rescorla–Wagner model (left) (related-samples $t$-test with Bonferroni correction for alternative models, $p < 0.001$). **f** The learning rate of the prospective Rescorla–Wagner model correlates significantly positively with participants' self-reported navigational abilities and preferences (SBSOD questionnaire[51]; Pearson correlation with Bonferroni correction for two tests, $p = 0.02$). Dots represent data from $n = 46$ participants; line represents linear regression line, with shaded regions as the 95% confidence interval. **a–c**, **e** Dots represent data from $n = 46$ participants ($n = 45$ in **c**); boxplots show median and upper/lower quartile with whiskers extending to the most extreme data point within 1.5 interquartile ranges above/below the quartiles; black circles with error bars correspond to mean ± SEM; distributions depict probability density functions of data points. Source data are provided as a Source Data file. \*\*\**p* < 0.001. All statistical tests were two-sided.

$r(44) = 0.11$, $p = 0.45$; with $\alpha = 0.025$, Bonferroni-corrected for two tests). In addition, we wondered whether the prospective component of integrating and extrapolating values over time in our task relates to model-based decision making in the two-stage task, which assesses reliance on a model of state transition probabilities across two decision stages[16]. Contrary to our expectations, we did not observe a significant correlation with model-based decision making in the two-stage task, potentially due to overall reduced model-based decision making in our sample (Supplementary Fig. 3a–c; correlation with learning rate: $r(44) = -0.19$, $p = 0.20$; correlation with overall performance: $r(44) = 0.21$, $p = 0.15$; with $\alpha = 0.025$, Bonferroni-corrected for two tests).

Taken together, our behavioral results demonstrate that participants were able to integrate and extrapolate changes along the two value dimensions of the space to guide choice, suggesting they formed a map of the relationships between options.

**Entorhinal cortex exhibits grid-like representation for value space**

Our behavioral results suggest that participants formed a relational value map. Relationships between landmarks in physical space, as well as non-spatial relational structures are represented by entorhinal grid cells in a cognitive map. We hypothesized that the

entorhinal cortex might also encode changing values using a grid-like representation. Such a neural representation would facilitate computations of directions of and distances between value changes over time and thereby enable efficient prediction of future values. Previous research has shown that the regular hexagonal firing pattern of grid cells in the entorhinal cortex translates to hexadirectional activity modulations during spatial navigation in fMRI[52]. In our prospective decision making task, a sequence of time points formed a trajectory through an underlying abstract value space (Fig. 1b). More specifically, participants moved along trajectories with directions ranging from 0°–350° in 10°-steps in each of the four task blocks (fMRI runs; Supplementary Fig. 1). If participants formed a cognitive map of changing values, akin to maps in physical space, then activity in the entorhinal cortex should show a hexadirectional modulation during this movement through the value space, with higher activity for trajectories aligned with the putative grid orientation (phase of the hexadirectional signal) than for trajectories misaligned with the putative grid orientation (Fig. 3a). To test this hypothesis, we implemented a cross-validation procedure, estimating the putative grid orientation using three of four task runs and testing for a hexadirectional modulation aligned to the orientation in the left-out run[52,53].

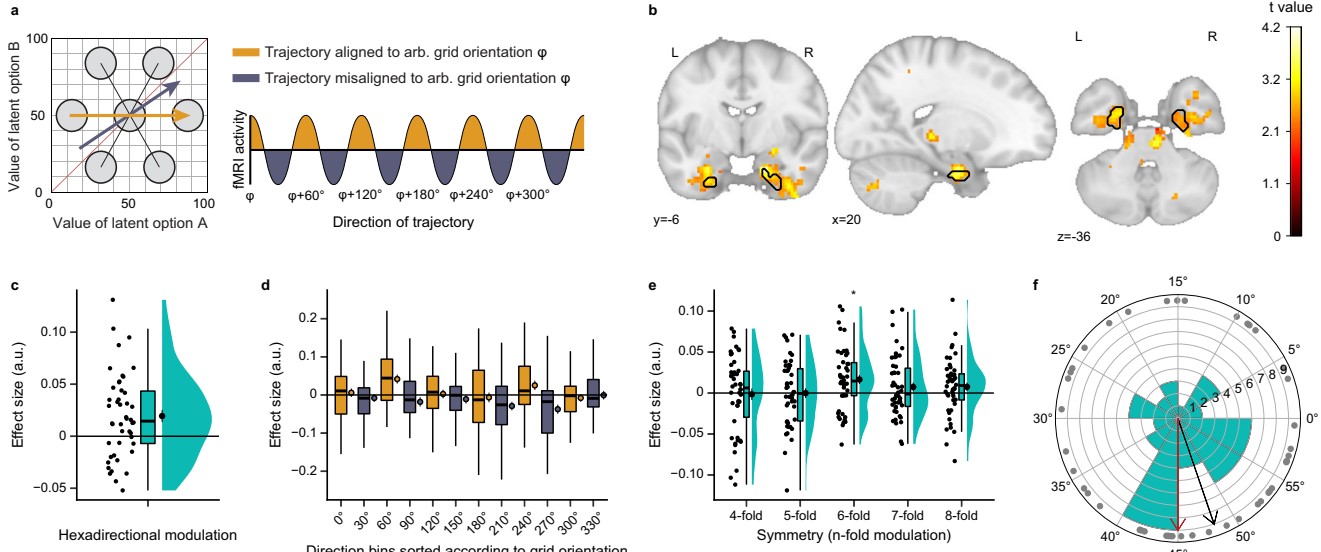

**Fig. 3 | Entorhinal cortex exhibits grid-like representation for value space.**
**a** fMRI hexadirectional analysis logic. Left: Schematic of a grid cell with an arbitrary orientation of φ = 0°, superimposed on the value space. Right: The regular hexagonal firing pattern of grid cells translates to hexadirectional activity modulations in fMRI, with higher activity for trajectories that are aligned (yellow) vs. misaligned (blue) with the grid orientation. **b** Grid-like hexadirectional modulation of activity, aligned with the putative entorhinal grid orientation. For visualization, statistical image is thresholded at $p_{uncorr} < 0.01$. Voxels within the black outline are significant after correction for multiple comparisons using small volume correction in the entorhinal cortex (one-sided non-parametric permutation test with TFCE and $p_{FWE} < 0.05$). Statistical image is displayed on the MNI template. **c** Visualization of the hexadirectional effect sizes in the significant entorhinal cluster in **b** across participants. **d** Visualization of the hexadirectional effect in the significant entorhinal cluster in **b** as effects of 30°-directional bins, sorted according to the putative entorhinal grid orientation. Yellow and blue depict aligned and misaligned directions, respectively (as in **a**). **e** Modulation of activity for different symmetries in the entorhinal cortex ROI (one-sided one-sample t-tests with Bonferroni correction; 6-fold refers to the hexadirectional modulation of interest, significant, $p = 0.003$; 4-fold ($p = 0.59$), 5-fold ($p = 0.51$), 7-fold ($p = 0.10$) and 8-fold ($p = 0.08$) refer to control symmetries and are n.s.). **f** Polar histogram of grid orientations of the significant entorhinal cluster in 60°-space across participants. Gray dots depict individual orientations from $n = 46$ participants, bars depict bins of 5°. Black arrow shows the circular mean of all participants' orientations. Red arrow highlights an orientation of 45°. Grid orientations cluster around 45° (V-test, $p = 0.01$). **c**–**e** Dots represent data from $n = 46$ participants; boxplots show median and upper/lower quartile with whiskers extending to the most extreme data point within 1.5 interquartile ranges above/below the quartiles; black circles with error bars correspond to mean ± SEM; distributions depict probability density functions of data points. Source data are provided as a Source Data file. *$p = 0.003$ with Bonferroni α = 0.01.

In line with our hypothesis, we observed significant hexadirectional modulation of activity in the entorhinal cortex (Fig. 3b, c; small volume correction with $p_{FWE} < 0.05$ TFCE; MNI peak voxel coordinates: 18,−6,−26; peak voxel $t(45) = 4.17$, $p_{FWE} = 0.003$; one-sided test). At the whole-brain level, we observed no further regions surviving correction (see Supplementary Table 1 for whole-brain results with a liberal threshold of $p_{uncorr} < 0.001$). To visualize the hexadirectional modulation in the significant entorhinal cluster, we sorted trajectories according to the putative grid orientation and illustrate effects of aligned and misaligned 30°-bins (Fig. 3d). A complementary ROI analysis of the entorhinal cortex confirmed the hexadirectional (6-fold) effect and showed that the modulation of activity was specific to a 6-fold symmetry in line with grid-like responses, as there were no significant effects for control symmetries (Fig. 3e; ROI analysis, one-sided tests with M ± SD: 4-fold $t(45) = −0.21$, $p = 0.59$, $−0.002 ± 0.05$ arb. units; 5-fold $t(45) = −0.02$, $p = 0.51$, $−0.0001 ± 0.04$ arb. units; 6-fold $t(45) = 2.91$, $p = 0.003$, $0.02 ± 0.04$ arb. units; 7-fold $t(45) = 1.34$, $p = 0.10$, $0.01 ± 0.04$ arb. units; 8-fold $t(45) = 1.42$, $p = 0.08$, $0.01 ± 0.04$ arb. units; control symmetries n.s.; with α = 0.01, Bonferroni-corrected for five tests). There was no significant correlation between the magnitude of hexadirectional modulation and task performance (Supplementary Fig. 4g; $r(44) = −0.08$, $p = 0.59$).

Furthermore, we performed exploratory analyses to investigate the relationship between the entorhinal grid system and the underlying value space. First, we wondered whether grid orientations would be anchored to a particular reference direction through the value space. We speculated that a direction of 45° constitutes a particularly informative reference direction because it indicates that values of both

options change at the same rate and—given that it is parallel to the 45°-diagonal of the value space—that there will be no switch of the more valuable option. In line with this idea, participants' performance was higher for trajectories with directions approximately parallel to the 45°-diagonal in switch trajectories (Supplementary Fig. 2f; interaction effect direction and switch: $F(2,86) = 7.18$, $p = 0.001$; post-hoc test parallel vs. perpendicular in switch trajectories: $t(43) = 2.90$, $p = 0.005$, with α = 0.008, Bonferroni-corrected for 6 pairwise tests; all other pairwise comparisons n.s. $p > 0.008$). We thus examined whether grid orientations in the significant entorhinal cluster would cluster around 45°, which was indeed the case (Fig. 3f; V-Test for mean orientation of 45° across participants: $p = 0.01$; circular M ± SD: 48.18° ± 15.99°). Secondly, we wondered whether the grid-like representation of the value space might be modulated by value (i.e., reward magnitude) itself. Recent evidence in rodents demonstrated restructuring of grid cells in response to reward locations during spatial navigation, with movement of grid fields towards reward locations and higher firing rates for grid fields closer to reward locations[54,55]. We, therefore, examined whether the magnitude of hexadirectional modulation differs between areas of the value space with relatively higher and lower values. To test this, we performed a median split of trajectories according to their mean value, i.e., contrasting trajectories in the lower left triangle of the space (low-value-area) with trajectories in the upper right triangle of the space (high-value-area). We note that this median split led to a substantial reduction of available trajectories per value condition and an unbalanced sampling of directions between the conditions, rendering this analysis less robust (Supplementary Fig. 4h; significant interaction between value condition and direction: $F(35,1575) = 3.56$,

$p < 0.001$). Using the significant entorhinal cluster as ROI, we then repeated the cross-validated hexadirectional analysis separately for the two value conditions. The analysis suggested no difference in hexadirectional modulation between the two conditions (Supplementary Fig. 4i; $t(45) = -1.30$, $p = 0.19$). However, it is interesting to note that—contrary to our expectations based on the rodent literature—it suggested a hexadirectional modulation effect in low-value-areas but not in high-value-areas (Supplementary Fig. 4i; low-value: $t(45) = 2.06$, $p = 0.02$; high-value: $t(45) = 0.30$, $p = 0.38$; one-sided tests).

Taken together, these results provide evidence that the entorhinal cortex encoded the abstract value space using a grid-like representation, suggesting the formation of a cognitive map.

### A network of brain regions tracks the prospective value difference during choices

To make a decision, representing values solely in a two-dimensional value map is not useful. Instead, values of the choice options also need to be mapped onto a single common scale for comparison. We thus tested whether neural signals track the value difference between the chosen and the unchosen option, especially in vmPFC based on previous literature. For this purpose, we modeled choice time points as a function of the chosen and unchosen values, derived from the prospective Rescorla-Wagner model, and contrasted these effects to test for a modulation by the value difference. We observed significant positive and negative modulation of neural activity by the value difference in a network of brain regions (see Fig. 4a for whole-brain effects; $p_{FWE} < 0.05$ TFCE-corrected; see Supplementary Table 2 for a list of significant clusters). Positive modulations reflected higher activity for a higher value difference and included amongst others vmPFC (MNI peak voxel coordinates: 3,42,−8; peak voxel statistics: $t(45) = 5.99$, $p_{FWE} < 0.001$), putamen, insular cortex, hippocampus, amygdala as well as motor and somatosensory cortex. Negative modulations reflected higher activity for a smaller value difference and included amongst others lateral parts of vPFC / OFC, dmPFC, thalamus and parietal cortex. These effects were still present when controlling for reaction time (Supplementary Fig. 5a) and when restricting the analysis to correct trials only (Supplementary Fig. 6a). Furthermore, the value difference effect in the vmPFC cluster correlated significantly positively with task performance (Fig. 4c, $r(44) = 0.34$, $p = 0.02$; after exclusion of outlier: $r(44) = 0.33$, $p = 0.03$).

Moreover, we aimed to investigate whether neural signals would track particularly the prospective component of the value difference, i.e., the difference based on the prospective values at the choice time point rather than the non-prospective values of the preceding time point. For this purpose, we subtracted value estimates of the original Rescorla-Wagner model (non-prospective) from value estimates of the prospective Rescorla-Wagner model, thereby extracting particularly the prospective value component for each option. We then modeled choice time points as a function of the prospective components of the chosen and the unchosen option and contrasted these effects to test for a modulation by the prospective value difference. Again, we observed widespread significant positive and negative modulation of neural activity by the prospective value difference (Fig. 4d; $p_{FWE} < 0.05$ TFCE-corrected; see Supplementary Table 3 for a list of significant clusters). Many clusters overlapped with those tracking the original value difference. However, a cluster in vmPFC/OFC extended more dorsally and bordered ACC (MNI peak voxel coordinates: −7,52,−8; peak voxel statistics: $t(44) = 5.75$, $p_{FWE} < 0.001$). This prefrontal cluster was still present when controlling for reaction time (Supplementary Fig. 5b), restricting the analysis to correct trials only (Supplementary Fig. 6b) and controlling for the distance between the choice location and the 45°-diagonal (Supplementary Fig. 7).

Taken together, these results demonstrate that a network of brain regions, including value regions such as vmPFC and dPFC, tracked not only the value difference between options during choices, but also particularly the prospective component of that value difference.

## Discussion

Our capacity to predict future values of choice options is central to many decisions we face in everyday life. Understanding the mechanisms by which the brain enables prospective decision making is therefore of particular importance. In this study, we combined fMRI with a prospective decision making task to investigate how the brain represents relational information about changing values of choice options in an abstract value space. Participants integrated and extrapolated changes along the two value dimensions to guide prospective choice. Crucially, while participants traversed the abstract value space along trajectories, the entorhinal cortex exhibited a grid-like representation, suggesting the formation of a cognitive map. A network of brain regions, including vmPFC and dPFC, tracked the prospective value difference between options.

Our finding of an entorhinal grid-like representation of an abstract value space dovetails with the broader idea of cognitive maps encoding abstract information[25,26,45,46] and research showing that vmPFC jointly encodes values and states[56]. Map-like representations of relationships between states enable prediction of future states[30]. In spatial navigation and memory, the hippocampal-entorhinal system is involved in prospective mental simulations and imaginations of events and navigational goals[57–62]. In the context of prospective value-based decision making, predicting future states corresponds to predicting future values of choice options, such as in the introductory example of a stock market. Critically, value changes over time can be conceptualized as sequences through an abstract value space, allowing for prospective decision making by facilitating the computation of geometric distances and directions. In line with this, it is noteworthy that the degree to which participants updated values over time in our value space task correlated with self-reported navigational abilities and preferences during spatial navigation in everyday life.

Our results extend recent evidence for map-like representations of value spaces in macaques into human research. For example, Bongioanni et al.[49] showed a grid-like representation in the macaque medial frontal cortex as a function of step-like transitions between static options in a space spanned by reward magnitude and probability. Knudsen and Wallis[50] found that hippocampal neurons in macaques, similar to place cells during spatial navigation in a physical space, encode position in a value space spanned by changing reward probabilities. Here, we provide evidence for an entorhinal grid-like representation of a value space during prospective decision making in humans.

Interestingly, our exploratory analysis suggests that the entorhinal grid system adapts to properties of the value space. More specifically, our results suggested an anchoring of grid orientations around 45° and participants' performance was also increased for directions parallel to 45° in switch trajectories. We speculate that a direction of 45° constitutes a particularly informative reference direction through our value space. This is because it indicates that values of both options change at the same rate and—given that it is parallel to the 45°-diagonal of the value space—that there will be no switch of the more valuable option. In line with this speculation, recent evidence in spatial navigation demonstrated anchoring of grid orientations to an informative axis in a virtual navigation arena which minimizes spatial uncertainty[63–66]. Our results suggest that grid orientations might anchor to an informative axis even in more abstract spaces. Furthermore, we wondered whether the grid-like representation of the value space might be modulated by value (i.e., reward magnitude) itself. Recent evidence in rodents demonstrated restructuring of grid cells in response to reward locations during spatial navigation, with movement of grid fields towards reward locations and higher firing rates for grid fields closer to reward locations[54,55]. While our results suggested no difference in the strength of grid-like representations between low-

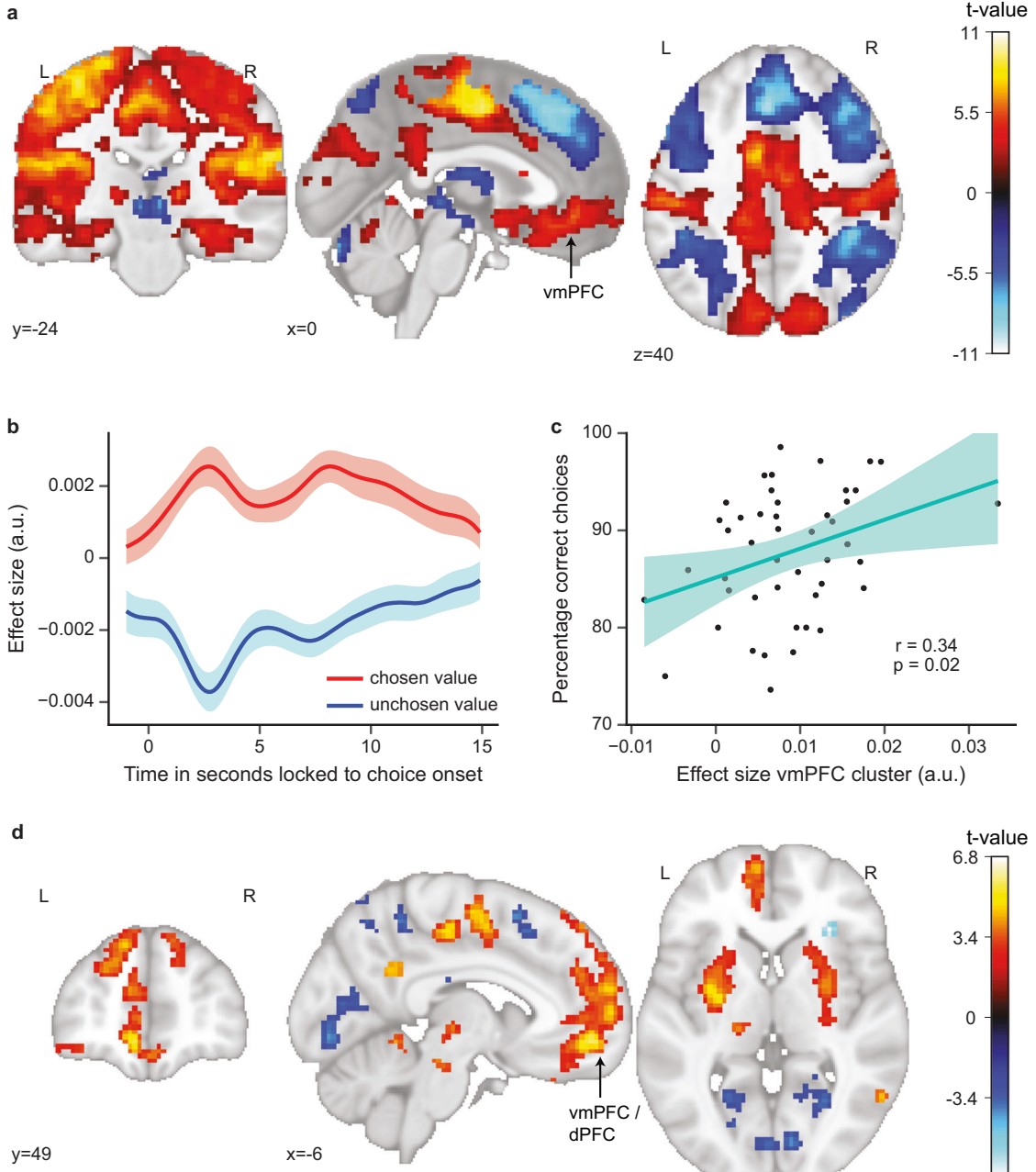

**Fig. 4 | A network of brain regions tracks the prospective value difference during choices. a** Modulation of activity by the difference between model-derived chosen vs. unchosen value during choices. Clusters depicted survive whole-brain correction (two-sided non-parametric permutation test with TFCE and $p_{FWE} < 0.05$). Statistical image is displayed on the MNI template. **b** Based on our expectation of a value difference effect in vmPFC, we visualize the effect in the vmPFC cluster by showing the time courses of the effect sizes of the chosen and unchosen value, time-locked to choice onset (choice onset at 0 s). Lines represent the mean across participants, with shaded regions as the 95% confidence interval. **c** The value difference effect in vmPFC correlates significantly positively with performance (two-sided Pearson correlation, $p = 0.02$). Dots represent data from $n = 46$ participants; line represents linear regression line, with shaded regions as the 95% confidence interval. **d** Modulation of activity by the prospective component of the value difference during choices. The prospective component refers to the influence of values estimated by the prospective Rescorla–Wagner model over values estimated by the original (non-prospective) Rescorla–Wagner model. Clusters depicted survive whole-brain correction (two-sided non-parametric permutation test with TFCE and $p_{FWE} < 0.05$). Statistical image is displayed on the MNI template. Source data are provided as a Source Data file.

and high-value areas of the value space, they surprisingly pointed towards a grid-like representation of the low-value but not the high-value area. We note that these effects of an exploratory analysis should be interpreted with caution, given the absence of a clear difference in the grid-like signal. One could speculate that participants' subjective gain of correct choices might have been higher in low-value than in high-value areas because the received reward in high-value areas was

high anyway, in agreement with notions of value distortions as value compression or diminishing utility[67,68]. While the main goal of our study was to assess whether a grid-like representation encodes a value space in principle, future studies could aim to investigate modulations of such a grid-like representation by value itself.

It is conceivable to represent values during our task by two separate number lines, without necessarily integrating them into a

two-dimensional space. Indeed, none of the participants reported having imagined the two-dimensional value space. However, our results, together with other studies demonstrating grid- and place-like representations of values[49,50], suggest indeed a neural representation of a two-dimensional space. Nevertheless, it is possible that different brain regions represent values differently, e.g., in a map-like format vs. combining them directly into a value difference or summary signal.

In light of this, it is interesting to note that we found no evidence for a grid-like representation in vmPFC, where previous studies also reported such grid-like representations[38,39,48,49]. While keeping in mind that it is difficult to interpret the lack of evidence, we can only speculate as to why this discrepancy might have arisen. Based on a body of literature implicating vmPFC in representing the one-dimensional value difference between options during decision making[1–11,69], one could assume that this is one of vmPFC's predominant coding schemes. It is conceivable that in our value-based decision making task with a high demand to track values, vmPFC engaged in its more prevalent coding scheme, which was encoding a one-dimensional value difference signal rather than the two-dimensional space. This idea is also in line with a recent study demonstrating a subjective value but no grid-like effect in vmPFC during a value-based intertemporal choice task[70]. In this case, different value representations in the hippocampal-entorhinal system and vmPFC could serve complementary functions. For example, while the entorhinal value map could support the prediction of future values by facilitating computations of directions of and distances between value changes over time, other brain regions such as vmPFC might read out the resulting values, map them onto a single common scale for comparison and thus generate a one-dimensional signal of the value difference used for decision making. In line with this notion and with previous literature in value-based decision making, a network of brain regions, including vmPFC, tracked the value difference between options during choices in our study. We observed both positive and negative modulations of activity by the value difference. While positive modulations might reflect the benefit of the chosen over the unchosen option, negative modulations could signal the relative value of the unchosen option as an alternative. Moreover, a vmPFC cluster extending more dorsally (dPFC) and bordering ACC tracked particularly the prospective component of the value difference. This prospective value difference effect is in line with reports of distinct model-based value correlates in dmPFC and reward rate tracking for trend-guided choice in neighboring dACC[14,17]. Furthermore, the pattern of a value difference signal directly relevant for choices in vmPFC and particularly prospective value components extending into dPFC dovetails with reports of a functional gradient, with vmPFC encoding values for executable choices and dmPFC encoding abstractly modeled values[71]. We speculate that the widespread involvement of brain regions in tracking values in our results might be explained by the high salience and relevance of values in the task. Ultimately, these value difference signals could be used to allocate attention to the more valuable option to guide eventual decision making in many foraging scenarios[14,72–74].

Finally, we would like to point to possible limitations of our study. First, the direction and rate of value changes were constant within a trial and differed across trials of our task, based on the sampling of different trajectories. Future studies could add noise to the value changes or vary them in a more fine-grained manner within trials to investigate how a map-like representation of a value space translates to more ecological scenarios. Secondly, we did not find evidence for a correlation between the grid-like representation and performance of the prospective decision making task across participants. In light of this, previous research reported mixed results for across-participant grid-behavior correlations, ranging from positive to negative to no reports of correlations[39,59,75]. Future studies could opt for testing such relationships more fine-grained on an individual participant level.

In conclusion, our results provide evidence that the human entorhinal cortex uses a grid-like representation to integrate relational information about changing values in an abstract value space during prospective decision making, suggesting the formation of a cognitive map. This map might be used to generate lower-dimensional signals of the value difference between options and their identities for choices. Thus, our findings provide novel insight for our understanding of cognitive maps as a mechanism to guide prospective decision making in humans.

## Methods
### Participants
51 participants took part in this study. The sample size was determined by a power analysis using G*Power[76]. This yielded a necessary sample size of 41 participants to achieve a statistical power of 80% for a small-to-medium effect size ($d = 0.4$, $\alpha = 0.05$, one-tailed $t$-test). Additionally, 10 participants were recruited to account for potential dropouts. All participants had normal or corrected-to-normal vision, no history of or current neurological or psychiatric disorders and were right-handed. Participants were recruited using the participant database of the Max Planck Institute for Human Cognitive and Brain Sciences, Leipzig, Germany.

For the data analysis, one participant was excluded due to missing fMRI data because of technical problems during data acquisition. Furthermore, four participants were excluded due to low performance of the prospective decision making task (performance criterion of 70% accuracy based on previous piloting). Thus, the final sample consisted of 46 participants (age: $M = 28.15$ years, $SD = 4.77$ years, range = 19–39 years; 25 female).

The study was approved by the ethics committee at the Medical Faculty at the University of Leipzig (421/19-ek) and all participants gave written informed consent prior to participation. Participants were reimbursed with a baseline fee of 10 € / h and could additionally earn a monetary bonus up to 10 € based on performance (see tasks for details).

### Experimental procedure
The study consisted of three parts and lasted approximately three hours in total. The first part took place in a behavioral laboratory (approx. 45 min). Here, participants received instructions and a training for the main task of the study, i.e., the prospective decision making task. In the second part (approx. 90 min), participants performed two tasks in the MRI scanner: First, they completed a picture viewing task (PVT) which served as an independent dataset to train a decoder for choice stimuli subsequently used in the prospective decision making task. Afterwards, they completed the prospective decision making task. In the third part (approx. 45 min), participants returned to a behavioral laboratory to complete two post-scanning tasks: the two-stage task[16] to study model-based vs. model-free decision making as well as the Santa Barbara Sense of Direction Scale (SBSOD) questionnaire[51] to assess navigational abilities and preferences.

### Task stimuli
Stimuli used for the picture viewing task and the prospective decision making task in the MRI scanner were category-specific pictures (faces, tools, scenes, body parts) which are known to elicit neural responses in category-selective regions of the occipital-temporal cortex. Stimuli were taken from publicly available stimulus datasets (faces: Righi et al.[77], Face images courtesy of Michael J. Tarr, Carnegie Mellon University, http://www.tarrlab.org/. Funding provided by NSF award 0339122; tools: Brady et al.[78], https://konklab.fas.harvard.edu/#; scenes: Konkle et al.[79], https://konklab.fas.harvard.edu/#; body parts: Cichy et al.[80], Kiani et al.[81], Kriegeskorte et al.[82], http://userpage.fu-berlin.de/rmcichy/fusion_project_page/main.html, https://www.cns.nyu.edu/kianilab/Datasets.html). From these stimulus sets, three

pictures of the categories faces, tools and scenes and one picture of the category body parts were pre-selected. From this preselection, one picture of each category was randomly chosen to create a set of four pictures for each participant.

## Prospective decision making task

Participants performed a prospective decision making task which required them to maximize reward by tracking and predicting values (i.e., reward magnitudes) associated with two choice options. The values of the two options changed over a sequence of time points and participants' goal was to choose the more valuable option at the next time point.

The options were represented by four category-specific stimuli (a face, a tool, a scene, a body part). Participants were instructed that two of these stimuli each formed a pair such that stimuli within a pair yielded the same value at a given time (they were value-congruent across the entire task). For example, the stimuli face and tool might have formed a pair and the stimuli scene and body part might have formed a pair. Hence, the task comprised two latent options (i.e., options A and B), with two value-congruent stimuli per option. Value congruencies between the four pictures were counterbalanced across participants.

Each trial consisted of an observation phase and an active choice. During the observation phase, participants viewed the two options along with their changing values over a sequence of time points (TP). Pictures of the two options were displayed on the left and right sides of the screen, with their associated current values indicated by numbers underneath. Across time points, two aspects changed: First, which of the two value-congruent stimuli of a given option was shown on the screen alternated each time point. Participants were instructed that when a stimulus and its current value were shown on the screen, the other stimulus of the pair currently yielded the same value. Secondly, the values of the two latent options changed over time points. Participants were instructed to carefully track these changes to be able to predict the options' values at the next time point. Each time point was presented for 2.5 s and was directly followed by the next time point. During the observation phase (initial time points), each option stayed on the same side of the screen to facilitate fast tracking of the changes. However, across trials the sides of the options were counterbalanced and distributed randomly. After 3–5 observed time points, only two pictures were presented and participants were asked to choose the option with the higher value at this future time point (choice time point). At the choice time point, the sides of the pictures on the screen (left / right) were random to prevent pure side-value associations. Participants were asked to indicate their choice by pressing the left or the right button on an MRI-compatible button box. Participants were given a maximum of 3 s to respond. After their choice (or the timeout), a fixation cross was presented at the center of the screen for an inter-stimulus interval sampled from a truncated exponential distribution (min = 3 s, max = 8 s, mu = 4 s, sampled mean = 4.1 s). Afterwards, a feedback screen was presented for 2.5 s, showing the pictures and their actual values at the choice time point. The value of the chosen option was highlighted in yellow. Lastly, a fixation cross was presented at the center of the screen for an inter-trial interval sampled from a truncated exponential distribution (min = 3 s, max = 8 s, mu = 4 s, sampled mean = 4.1 s).

The task comprised 144 trials. Half of the trials involved a choice as described above, with choices at the 4th, 5th or 6th time point. The other half of the trials proceeded without any choice and consisted of six time points (passive trials). The purpose of including longer trials without any choice was to improve the estimation of trajectory-related fMRI signals for the planned analysis of hexadirectional signals (grid-like representation, see below). Choice and passive trials were inter-mixed randomly so that participants would always need to track the values in a given trial and be ready to indicate their choice.

Crucially, a sequence of time points in a trial formed a trajectory through an underlying abstract two-dimensional value space. The two dimensions of the space corresponded to the values associated with the two options (ranging from 0 to 100). Each time point with its current values of the two options corresponded to a particular location in the value space and a trial could therefore be conceptualized as movement along a trajectory through the space. Trajectories were sampled with directions (angles) ranging from 0°–350° in 10°-steps (0° referring to a value increase along the x-dimension of the space but no change in the y-dimension).

In this space, the 45°-diagonal represented locations where the two options had the same values. Trajectories crossing the 45°-diagonal therefore involved a switch in which of the two options was more valuable. Half of all trajectories in the task involved a switch while the other half did not (switch vs. non-switch trajectories). The switch time point was defined as the first time point after the 45°-diagonal. The switch time point appeared equally often at the 4th, 5th and 6th time point across all switch trajectories (equal sampling both in choice and in passive trials / trajectories).

Time points (locations) along a trajectory were sampled equidistantly, i.e., the distance between two consecutive time points was the same within a given trajectory. The values shown to participants during the task were rounded to integers. Depending on the direction (angle) of a trajectory, rounding could lead to differences of +/−1 in value changes between time points but it was ensured that this would not change the identity of the more valuable option at choice time points. Furthermore, the task included two types of trajectories with regard to the distance between two consecutive time points: trajectories with a relatively smaller distance of 6 (referred to as short-distance trajectories) and trajectories with a relatively larger distance of 10 (referred to as long-distance trajectories). For each distance type, one set of trajectories (36 directions, 0°–350°) was realized as choice trials and one set of trajectories as passive trials. Furthermore, for a given distance type each direction was once realized as a switch trajectory and once as a non-switch trajectory. The assignment of switch vs. non-switch to choice vs. passive trials was pseudorandom with the condition that in choice trials, each direction was realized as a switch trajectory at least once across both distance types. This ensured that a response from the participant was sampled for all directions 0°–350° involving a switch.

As noted above, choices occurred either at the 4th, 5th or 6th time point. In non-switch trajectories, the 4th, 5th and 6th time point equally often constituted the choice time point for each distance type. In switch trajectories, for each distance type half of the choice time points sampled the switch time point (18 trials). The other half sampled the time point before the switch (pre) and the time point after the switch (post) equally often (i.e., 50% switch time point, 25% pre time point, 25% post time point; note that for 6 participants at the beginning of the study the balance between pre and post differed up to +/−3 trials).

The total of 144 trials (trajectories) was pseudorandomly distributed over four task blocks (fMRI runs) so that each block:

1. sampled all 36 directions ranging from 0°–350° in 10°-steps (hence, 36 trials per block),
2. sampled switch and non-switch trajectories equally often,
3. sampled choice and passive trajectories equally often,
4. sampled the switch time point as the choice time point equally often (one trial more in one block),
5. sampled short- and long-distance trajectories equally often,
6. and for each distance type sampled choice and passive trajectories equally often.

In each block, the order of trajectories was randomized according to the direction.

In each block, trajectories were positioned within the circle created by a radius of 50 from the central point of the two-dimensional

value space (coordinates: x = 50, y = 50). Equal positioning of trajectories in the relevant circular area of the space was achieved by a genetic algorithm. Its goal was to position trajectories so as to minimize the standard deviation of the number of time points (locations on trajectories) falling into the $10 \times 10$ sub-squares of the relevant circular area (for 2000 generations).

Each block lasted approx. 13 min ($M = 12.74$ min, $SD = 0.15$ min). It started with a fixation cross presented at the center of the screen for 10 s before the first trial. After the last trial, a fixation cross was presented at the center of the screen for 15 s, followed by a message informing the experimenter that the block finished and the MRI run could be stopped. After each block, participants received feedback about their performance in the given block. The feedback stated the number of correctly answered trials as well as the earned monetary bonus in the given block. More specifically, in each block a trial was randomly chosen for the bonus. If the answer in this trial was correct, the highest values across the entire trial were summed up and converted into a bonus (so that a value of 50 yielded 0.20 €). If the answer in this trial was false, no bonus was won. Participants were instructed about this bonus beforehand. Participants could take a short break before the next block.

Participants received instructions for the task and a training of 25 trials before performing the main task in the scanner. During training, incorrectly answered trials were repeated until answered correctly. For instructions and training only, a distinct set of stimuli of everyday objects from a publicly available stimulus dataset[78] was used.

At the end of the study, participants were asked which strategies they used to solve the task. Then they were told about the underlying two-dimensional value space and asked whether they imagined such a space. The first question about the strategies was only verbal so that we cannot exactly quantify them. While participants mentioned, amongst others, imagining separate number lines and trying to calculate the value changes, they also reported that they felt that the value and stimulus changes happened very fast. For the second question—whether participants imagined the underlying two-dimensional value space—we recorded answers to potentially exclude those participants from the analysis. No participant reported having imagined the underlying two-dimensional value space.

The task was programmed in Python 3.7 using the PsychoPy package[83] (version 3.1.5; https://lindeloev.net/psychopy-course/) in Spyder (https://www.spyder-ide.org/; version 4.0.0b3) distributed via Anaconda (https://www.anaconda.com/; version 2019.03). The instruction was programmed using the PsychoPy Builder[83] (version 2020.2.3).

## Picture viewing task (PVT)
Before the prospective decision making task, participants performed a picture viewing task (PVT) which served as an independent dataset to train a decoder for subsequent analyses. Participants viewed a stream of pictures of the category-specific stimuli which were later used as choice stimuli in the prospective decision making task. The PVT was participants' first exposure to these category-specific stimuli during the study.

To ensure that participants paid attention to the presentation of the stimuli, they performed a one-back cover task. In each trial, a stimulus was presented for 2 s at the center of the screen. This was followed by a fixation cross at the center of the screen for an inter-trial interval sampled from a truncated exponential distribution (min = 2 s, max = 8 s, mu = 3 s, sampled mean = 3.3 s). If the fixation cross was red, participants had to judge whether the stimulus in the next trial was the same as the preceding stimulus before the fixation cross (test trial). If the fixation cross was white, no judgement was required (regular trial). In test trials, participants had to indicate their judgement by pressing one of two buttons on an MRI-compatible button box if the stimulus was the same as the preceding one and the other button if it was

different. Button contingencies (left vs. right button for which type of judgement) were counterbalanced and randomized across participants. Participants were instructed to press the button while the stimulus was presented (hence maximum response time of 2 s). After a test trial the task proceeded without direct trial-specific feedback.

The task consisted of 65 trials, with 14 regular trials per stimulus (+1 for one stimulus) and 2 test trials per stimulus. The sequence of trials was generated pseudorandomly so that every stimulus was preceded equally often by every other stimulus including self-repetitions (i.e., serial-order counterbalanced sequence)[84]. Test trials were distributed pseudorandomly over the trial sequence so that every bin of 8 trials contained a test trial. Of the 2 test trials per stimulus, one trial was realized as a self-repetition trial (same-stimulus-judgement) and one as a non-self-repetition trial (different-stimulus-judgement).

The task lasted approx. 6 min ($M = 6.18$ min, $SD = 0.02$ min). The task started with a fixation cross presented at the center of the screen for 10 s before the first trial. After the last trial, a fixation cross was presented at the center of the screen for 15 s, followed by a message informing the experimenter that the task finished and the MRI run could be stopped. Afterwards, participants received feedback about their task performance. The feedback stated the number of correctly answered trials as well as the earned monetary bonus. Participants were instructed that for each correctly answered test trial they would earn a bonus of 0.15 €. Participants could take a short break after the task.

Serial-order counterbalancing of the trial sequence was performed in Matlab using a script by Brooks[84]. The task was programmed in Python 3.7 using the PsychoPy package[83] (version 3.1.5; https://lindeloev.net/psychopy-course/) in Spyder (https://www.spyder-ide.org/; version 4.0.0b3) distributed via Anaconda (https://www.anaconda.com/; version 2019.03).

## Two-stage task
Participants performed the two-stage decision making task developed by Daw et al.[16] to study model-based vs. model-free decision making. The task structure consisted of two decision stages and participants' goal was to maximize rewards obtained by decisions at the second stage. Stimuli were character symbols.

In each trial, decisions were made at two stages. At the first stage, participants had to choose between two stimuli by pressing one of two buttons on a keyboard. The chosen stimulus moved to the top of the screen. Below, one of two second-stage states was presented. The second stage consisted of two other stimuli and participants had to choose one of them by pressing one of two buttons on a keyboard. The second-stage decision was either rewarded (displayed by a coin) or not (displayed by a red X), presented on the screen for 1 s. Participants had a maximum of 3 s to indicate their decision.

Transitions to the two second-stage states depended probabilistically on the first-stage decision. One stimulus at the first stage led to one second-stage state with a higher probability of 70% (common transition) while it led to the other second-stage state with a lower probability of 30% (rare transition). This transition pattern was reversed for the other first-stage stimulus. At the second stage, reward probabilities were determined by a Gaussian process with a standard deviation of 0.025 and reflecting boundaries of 0.25 and 0.75.

Participants were instructed that one of the first-stage stimuli primarily lead to one second-stage state and vice versa and that this pattern would remain constant across the task. Furthermore, they were instructed that reward probabilities of second-stage stimuli could change and that collected rewards would be translated into a monetary bonus at the end of the task.

Participants performed a training with 50 trials and a distinct set of stimuli. The main task comprised 201 trials. At the end of the task, participants received feedback stating the earned monetary bonus based on their performance (bonus was calculated as 0.015 € per obtained reward). The task lasted approx. 30 min.

For this task, a PsychoPy-based Python script from a publicly available repository (Abraham Nunes, https://abrahamnunes.github.io/paradigms/) was used. The script was adapted in Python 3.7 using the PsychoPy package[83] (version 3.1.5; https://lindeloev.net/psychopy-course/) in Spyder (https://www.spyder-ide.org/; version 4.0.0b3) distributed via Anaconda (https://www.anaconda.com/; version 2019.03).

## Santa Barbara Sense of Direction Scale (SBSOD)

Participants filled out the Santa Barbara Sense of Direction Scale (SBSOD) questionnaire[51] on a computer. This questionnaire measures navigational abilities and preferences and consists of 15 items (self-referential statements). Items were presented subsequently in the upper part of the screen, together with a 7-point rating scale underneath (1 = strongly agree, 7 = strongly disagree). Participants were asked to indicate their response by pressing the respective number on a keyboard and confirming their response with enter (response was self-paced).

To compute the score of the questionnaire, responses to positive items were reverse-coded so that a higher overall score reflected higher navigational abilities and preferences.

The task was programmed in Python 3.7 using the PsychoPy package[83] (version 3.1.5; https://lindeloev.net/psychopy-course/) in Spyder (https://www.spyder-ide.org/; version 4.0.0b3) distributed via Anaconda (https://www.anaconda.com/; version 2019.03).

## MRI data acquisition

MRI data were recorded using a 3 Tesla Siemens Magnetom Prisma Fit scanner (Siemens, Erlangen, Germany) with a 32-channel head coil.

After a localizer scan, functional scans (fMRI) for the picture viewing task and the four runs of the prospective decision making task were acquired using T2\*-weighted whole-brain gradient-echo echo planar imaging (GE-EPI) with multiband acceleration, sensitive to blood-oxygen-level-dependent (BOLD) contrast[85,86]. Settings of the fMRI sequence were as follows: TR = 1500 ms; TE = 22 ms; voxel size = 2.5 mm isotropic; field of view = 204 mm; flip angle = 70°; partial fourier = 0.75; bandwidth = 1794 Hz/Px; multi-band acceleration factor = 3; 69 slices interleaved; distance factor = 0%; phase encoding direction = A-P. On average, 253 volumes were recorded for the PVT ($M$ = 252.76 volumes, $SD$ = 5.45 volumes) and 514 volumes per run of the prospective decision making task ($M$ = 513.70 volumes, $SD$ = 7.57 volumes).

After the second run of the prospective decision making task, field maps were acquired to measure and later correct for magnetic field inhomogeneities. Field maps were acquired using both opposite phase-encoded EPIs and a double echo sequence. Settings of the opposite phase-encoded EPIs were as follows: TR = 8000 ms; TE = 50 ms; voxel size = 2.5 mm isotropic; field of view = 204 mm; flip angle = 90°; partial fourier = 0.75; bandwidth = 1794 Hz/Px; multi-band acceleration factor = 1; 69 slices interleaved; distance factor = 0%. Settings of the double echo sequence were as follows: TR = 620 ms; TE1 = 4.00 ms; TE2 = 6.46 ms; voxel size = 2.5 mm isotropic; field of view = 204 mm; flip angle = 60°; bandwidth = 412 Hz/Px; 69 slices interleaved; distance factor = 0%.

At the end of the scanning session, a T1-weighted MPRAGE anatomical scan was acquired (TR = 2300 ms; TE = 2.98 ms; voxel size = 1 mm isotropic; field of view = 256 mm; flip angle = 9°; bandwidth = 240 Hz/Px; distance factor = 50%).

To measure physiological noise signals during the fMRI runs, pulse oximeter data were recorded on participants' hands using a Siemens pulse sensor and the PhysioLog function of the multiband sequence.

Task stimuli were projected on a screen via a mirror attached to the head coil and behavioral responses were collected with an MRI-compatible button box.

## Behavioral data analysis software

We performed all behavioral analyses in Python 3.8 using Spyder (https://www.spyder-ide.org/; version 5.1.5) distributed via Anaconda (https://www.anaconda.com/; version 2020.11). Statistical analyses were based on the packages scipy (version 1.10.0) and statsmodels (version 0.13.2). $T$-tests and correlations tests were based on non-parametric permutation-based approaches to assess significance (10000 permutations). If not stated otherwise, we used an alpha level of .05 and two-sided tests.

## Performance and reaction time analysis of the prospective decision making task

We calculated performance in the prospective decision making task as the proportion of trials with a correct choice, defined as choice of the objectively more valuable option at the choice time point. We first assessed whether participants met our performance criterion of at least 70% (based on previous piloting, see Participants) to be included in the final analysis sample. For this purpose, we left trials with missing responses labeled as incorrect (total of 50 trials with missing responses across participants). For further analyses, we labeled trials with missing responses as NaNs so that they were not considered in the analyses. Furthermore, we labeled trial scores as NaNs if both options had the same objective value at the choice time point (same value could happen due to constraints by the direction (angle) of the trajectory; $M$ = 2.2 trials, $SD$ = 1.19 trials across participants). We log-transformed reaction times.

In switch trajectories, we tested whether performance at the switch time point was better than expected by chance using a one-sample $t$-test against 50% (and as controls also for the pre and post time point). Furthermore, we tested whether the time point in switch trajectories (pre, switch, post) influenced performance and reaction times using repeated measures ANOVAs and post-hoc pairwise tests (related-samples $t$-tests, with α = 0.016, Bonferroni-corrected for three comparisons).

To estimate the effect of the distance between the choice location and the 45°-diagonal of the value space on performance, we implemented a logistic regression for each participant predicting trial scores based on the distance. We then tested participant-specific effect sizes against 0 using a one-sample $t$-test on the group level. As a control, we repeated this analysis using only choices in switch trajectories where locations lay inherently closer to the diagonal. In both all and switch-trajectories-only analyses, one extreme outlier data point was excluded from the group level test (data point was 301.08 SD and 128.28 SD away from sample mean without that data point). We visualized the effect by showing correct and incorrect choice locations in the value space. For reaction times, we tested the effect of the distance from the diagonal using participant-specific linear regressions.

To test whether performance was influenced by short- vs. long-distance trajectories, we implemented a repeated measures ANOVA with the factors distance type and time point (pre, switch, post).

We also analyzed performance for different directions of trajectories. We binned directions according to quadrants, reflecting whether values increased or decreased for both options or in opposite directions (Q1: 10–80°, Q2: 100–170°, Q3: 190–260°, Q4: 280–350° and cardinal directions of 0°, 90°, 180° and 270° as a separate bin). We tested whether the quadrant influenced performance using a repeated measures ANOVA. Furthermore, we compared performance for directions approximately parallel to the 45°-diagonal (sampled directions: 40°, 50°, 220°, 230°), directions approximately perpendicular to the 45°-diagonal (sampled directions: 130°, 140°, 310°, 320°) and all other directions, by taking into account possible differences due to the differences in switches, using a repeated measures ANOVA with the within-subject factors direction and switch vs. non-switch trajectory (two participants excluded due to missing data for some conditions). We further investigated the interaction effect using post-hoc pairwise

related-samples $t$-tests with Bonferroni correction ($\alpha = 0.008$, Bonferroni-corrected for 6 pairwise tests).

## Reinforcement learning model for the prospective decision making task

We investigated whether a reinforcement learning model which captured the prospective nature of the task, i.e., the value changes over time, fitted participants' choice behavior better than a model that did not. To this end, we modified a Rescorla-Wagner model[12]. The Rescorla-Wagner model updates value estimates of choice options according to a prediction error, defined as the difference between the expected value and the received outcome. We modified the original Rescorla–Wagner model so that it updated value estimates within a trial based on prediction errors and additionally value changes over time points. We refer to this modified version as the prospective Rescorla-Wagner model. More specifically, each option's value within a trial was updated according to:

$$V_{TP+1} = V_{TP} + \alpha^*(O_{TP} + C_{TP} - V_{TP}) \text{ with } C_{TP} = O_{TP} - O_{TP-1}, \quad (3)$$

whereby $V_{TP}$ and $V_{TP+1}$ are values at the current and next time points, respectively, $O_{TP}$ is the outcome at the current time point, $C_{TP}$ reflects how the value has changed from the previous to the current time point and $\alpha$ is the learning rate (free parameter of the model). Value estimates of both options were translated into choices by computing the probability of each option's choice using a softmax function:

$$P_A = \frac{e^{\beta^* V_A}}{e^{\beta^* V_A} + e^{\beta^* V_B}}, \quad (4)$$

with $P_A$ as the probability of choosing option A, $e$ as the exponential, $V_A$ and $V_B$ as the values of options A and B (values divided by 100) and $\beta$ as inverse temperature indicating the determinacy of choices (free parameter of the model). In each trial, values were initialized with the objective values of the first time point, outcomes of the second time point were received and value predictions were made for the following time points.

We fitted this prospective Rescorla–Wagner model to each participant's choice data and searched for the best-fitting estimates of the free parameters α and β by minimizing the negative log-likelihood of the model. Parameter estimates were initially bound to ranges [0,1] for α and [0,100] for β. As we observed a ceiling effect for α, we removed its upper bound to allow estimates greater than 1.

We compared the fit of this prospective Rescorla–Wagner model to the fit of the original Rescorla–Wagner model. The original Rescorla–Wagner model does not consider value changes over time points (no prospective component):

$$V_{TP+1} = V_{TP} + \alpha^*(O_{TP} - V_{TP}). \quad (5)$$

Notations, translation of value estimates into choice probabilities using a softmax function and model fitting were the same as described above. Parameter estimates of the original Rescorla-Wagner model were bound to ranges [0,1] for α and [0,100] for β. We compared the fits of the prospective and the original Rescorla–Wagner model by testing for a difference in the Akaike Information Criterion (AIC) using a related-samples $t$-test (with $\alpha = 0.01$, Bonferroni-corrected for five tests including alternative models, see below). We extracted parameter estimates of the winning model (prospective Rescorla-Wagner model). As a control, we correlated the learning rate α with performance at the switch and the pre time point using Pearson correlations.

In addition to the prospective Rescorla–Wagner model described above, we implemented four alternative control models which similarly aimed to capture the prospective nature of the task:

- Prospective control model 1: Similar to prospective Rescorla-Wagner model described above, but $C_{TP}$ as the option's value change is updated itself across time points with its own learning rate:

$$V_{TP+1} = V_{TP} + \alpha^*(O_{TP} + C_{TP\_exp} - V_{TP}) \quad (6)$$

with $C_{TP\_exp} = C_{TP}$ and $C_{TP} = O_{TP} \cdot O_{TP-1}$ for the first update within a trial, and $C_{TP\_exp} = C_{TP\_exp} + \alpha_C^* C_{TP}$ afterwards.

- Prospective control model 2: Value update with standard prediction error and an additional parameter for the value change:

$$V_{TP+1} = V_{TP} + \alpha^*(O_{TP} - V_{TP}) + \delta^*(O_{TP} - O_{TP-1}) \quad (7)$$

- Prospective control model 3: Value update with standard prediction error and expected prediction error, similar to expected prediction error models in Wittmann et al.[14]:

$$V_{TP+1} = V_{TP} + \alpha^* PE + PE_{exp} \quad (8)$$

with $PE = O_{TP} \cdot V_{TP}$ and $PE_{exp} = PE$ for the first update within a trial, and $PE_{exp} = PE_{exp} + \alpha^*(PE \cdot PE_{exp})$ afterwards

- Prospective control model 4: Similar to prospective control model 3, but the expected prediction error is updated with its own learning rate:

$$V_{TP+1} = V_{TP} + \alpha^* PE + PE_{exp} \quad (9)$$

with $PE = O_{TP} \cdot V_{TP}$ and $PE_{exp} = PE$ for the first update within a trial, and $PE_{exp} = PE_{exp} + \alpha_{PEE}^*(PE \cdot PE_{exp})$ afterwards.

To allow similar parameter fits as for the prospective Rescorla-Wagner model described above, we removed the upper bound of 1 for learning rates of these control models. The prospective Rescorla–Wagner model described above fitted the data better than any of the control models (test for difference in AIC using related-samples $t$-tests; PC1: $t(45) = -3.58$, $p < 0.001$; PC2: $t(45) = -4.30$, $p < 0.001$; PC3: $t(45) = -6.93$, $p < 0.001$; PC4: $t(45) = -5.92$, $p < 0.001$, with $\alpha = 0.0125$, Bonferroni-corrected for four comparisons).

## Performance analysis of picture viewing task

We calculated performance in the one-back cover task of the picture viewing task as the proportion of correctly answered test trials. For this purpose, we labeled trials with missing responses as incorrect. For two participants at the beginning of the study, button presses were not registered due to a technical mistake (except for the first test trial). Therefore, we could not assess performance for these participants. However, we still used their fMRI data of the picture viewing task for the fMRI analysis as the purpose of the one-back cover task was only to ensure participants' attention to the stimuli. In addition, two participants reported that they confused the buttons for the two response types (button contingencies: left or right button for same or different stimulus judgement). Indeed, their responses matched exactly the opposite pattern of all correct trial-wise responses. For this reason, we reverse-coded their responses to calculate their performance.

## Analysis of model-based decision making in the two-stage task

Analogously to Daw et al.[16], we tested whether the probability of repeating a first-stage choice depended on the reward and the transition type in the preceding trial. For this purpose, we labeled each trial as 1 if participants chose the same first-stage stimulus as in the preceding trial and as 0 if not. We calculated stay percentages for the factors reward (received or not) and transition type (common or rare). Across participants, we tested whether stay percentages were

influenced by reward, transition type and their interaction using a repeated measures ANOVA.

In addition, we fitted each participant's choice data using the hybrid reinforcement learning model as described in Daw et al.[16]. This model learns values by both model-based and model-free decision algorithms. Both values are weighted by a free parameter indicating the influence of model-based values on choices (ranging from 0 for model-free to 1 for model-based). In addition, the model contains separate learning rates and inverse temperatures for the two stages as well as a perseverance parameter and an eligibility trace. We fitted this model using its implementation in the hBayesDM package[87] (version 1.1.1; model: ts_par7).

We correlated estimates of the model-based parameter of the two-stage task with the learning rate and overall performance of the prospective decision making task using Pearson correlations (with $\alpha = 0.025$, Bonferroni-corrected for two tests).

### Santa Barbara Sense of Direction Scale (SBSOD) correlations
We correlated scores of the SBSOD with the learning rate and overall performance of the prospective decision making task using Pearson correlations (with $\alpha = 0.025$, Bonferroni-corrected for two tests).

### MRI analysis software
We performed all MRI analyses (preprocessing and main analyses) in Python 3.8 using Spyder (https://www.spyder-ide.org/; version 5.1.5) distributed via Anaconda (https://www.anaconda.com/; version 2020.11). MRI analyses were mainly based on the packages nilearn (version 0.9.0), nibabel (version 3.2.1), scikit-learn (version 1.0.1) as well as FSL (version 6.0.3), ANTS (version 2.3.5) and tools stated below. Statistical analyses were based on the package scipy (version 1.10.0) and statsmodels (version 0.13.2). T-tests and correlations tests were based on non-parametric permutation-based approaches to assess significance (10000 permutations). If not stated otherwise, we used an alpha level of 0.05 and two-sided tests.

### Conversion of MRI data to the Brain Imaging Data Structure (BIDS) standard
We converted DICOM files of the MRI scanner to NIfTI files and reorganized them according to the BIDS standard[88] using the tool dcm2bids (version 2.1.6, https://unfmontreal.github.io/Dcm2Bids/). Furthermore, we removed facial structure in the anatomical scan using the tool pydeface (version 2.0.0, https://github.com/poldracklab/pydeface) to further anonymize the data.

### Preprocessing by fMRIPrep
Results included in this manuscript come from preprocessing performed using fMRIPrep 20.2.6[89,90] (RRID:SCR_016216), which is based on Nipype 1.7.0[91,92] (RRID:SCR_002502).

### Anatomical data preprocessing
A total of 1 T1-weighted (T1w) images were found within the input BIDS dataset. The T1-weighted (T1w) image was corrected for intensity non-uniformity (INU) with N4BiasFieldCorrection[93], distributed with ANTs 2.3.3[94] (RRID:SCR_004757), and used as T1w-reference throughout the workflow. The T1w-reference was then skull-stripped with a Nipype implementation of the antsBrainExtraction.sh workflow (from ANTs), using OASIS30ANTs as target template. Brain tissue segmentation of cerebrospinal fluid (CSF), white-matter (WM) and gray-matter (GM) was performed on the brain-extracted T1w using fast (FSL 5.0.9, RRID:SCR_002823[95]). Brain surfaces were reconstructed using recon-all (FreeSurfer 6.0.1, RRID:SCR_001847[96]), and the brain mask estimated previously was refined with a custom variation of the method to reconcile ANTs-derived and FreeSurfer-derived segmentations of the cortical gray-matter of Mindboggle (RRID:SCR_002438[97]). Volume-based spatial normalization to two standard spaces (MNI152NLin2009cAsym, MNI152NLin6Asym) was performed through nonlinear registration with antsRegistration (ANTs 2.3.3), using brain-extracted versions of both T1w reference and the T1w template. The following templates were selected for spatial normalization: *ICBM 152 Nonlinear Asymmetrical template version 2009c*[98] [RRID:SCR_008796; TemplateFlow ID: MNI152NLin2009cAsym], *FSL's MNI ICBM 152 nonlinear 6th Generation Asymmetric Average Brain Stereotaxic Registration Model*[99] [RRID:SCR_002823; TemplateFlow ID: MNI152NLin6Asym],

### Functional data preprocessing
For each of the 5 BOLD runs found per subject (across all tasks and sessions), the following preprocessing was performed. First, a reference volume and its skull-stripped version were generated using a custom methodology of *fMRIPrep*. A B0-nonuniformity map (or *fieldmap*) was estimated based on two (or more) echo-planar imaging (EPI) references with opposing phase-encoding directions, with 3dQwarp[100] (AFNI 20160207). Based on the estimated susceptibility distortion, a corrected EPI (echo-planar imaging) reference was calculated for a more accurate co-registration with the anatomical reference. The BOLD reference was then co-registered to the T1w reference using bbregister (FreeSurfer) which implements boundary-based registration[101]. Co-registration was configured with six degrees of freedom. Head-motion parameters with respect to the BOLD reference (transformation matrices, and six corresponding rotation and translation parameters) are estimated before any spatiotemporal filtering using mcflirt (FSL 5.0.9[102]). BOLD runs were slice-time corrected to 0.708 s (0.5 of slice acquisition range 0–1.42 s) using 3dTshift from AFNI 20160207 (ref. 100, RRID:SCR_005927). The BOLD time-series were resampled onto the following surfaces (FreeSurfer reconstruction nomenclature): *fsnative*, *fsaverage*. The BOLD time-series (including slice-timing correction when applied) were resampled onto their original, native space by applying a single, composite transform to correct for head-motion and susceptibility distortions. These resampled BOLD time-series will be referred to as *preprocessed BOLD in original space*, or just *preprocessed BOLD*. The BOLD time-series were resampled into standard space, generating a *preprocessed BOLD run in MNI152NLin2009cAsym space*. First, a reference volume and its skull-stripped version were generated using a custom methodology of *fMRIPrep*. Automatic removal of motion artifacts using independent component analysis (ICA-AROMA[103]) was performed on the *preprocessed BOLD on MNI space* time-series after removal of non-steady state volumes and spatial smoothing with an isotropic, Gaussian kernel of 6 mm FWHM (full-width half-maximum). Corresponding "non-aggresively" denoised runs were produced after such smoothing. Additionally, the "aggressive" noise-regressors were collected and placed in the corresponding confounds file. Several confounding time-series were calculated based on the *preprocessed BOLD*: framewise displacement (FD), DVARS and three region-wise global signals. FD was computed using two formulations following Power (absolute sum of relative motions[104], and Jenkinson (relative root mean square displacement between affines[102]). FD and DVARS are calculated for each functional run, both using their implementations in *Nipype* (following the definitions by ref. 104). The three global signals are extracted within the CSF, the WM, and the whole-brain masks. Additionally, a set of physiological regressors were extracted to allow for component-based noise correction (*CompCor*[105]). Principal components are estimated after high-pass filtering the *preprocessed BOLD* time-series (using a discrete cosine filter with 128 s cut-off) for the two *CompCor* variants: temporal (tCompCor) and anatomical (aCompCor). tComp-Cor components are then calculated from the top 2% variable voxels within the brain mask. For aCompCor, three probabilistic masks (CSF, WM and combined CSF + WM) are generated in anatomical space. The implementation differs from that of Behzadi et al.[105] in that instead of eroding the masks by 2 pixels on BOLD space, the aCompCor masks

are subtracted a mask of pixels that likely contain a volume fraction of GM. This mask is obtained by dilating a GM mask extracted from the FreeSurfer's *aseg* segmentation, and it ensures components are not extracted from voxels containing a minimal fraction of GM. Finally, these masks are resampled into BOLD space and binarized by thresholding at 0.99 (as in the original implementation). Components are also calculated separately within the WM and CSF masks. For each CompCor decomposition, the $k$ components with the largest singular values are retained, such that the retained components' time series are sufficient to explain 50 percent of variance across the nuisance mask (CSF, WM, combined, or temporal). The remaining components are dropped from consideration. The head-motion estimates calculated in the correction step were also placed within the corresponding confounds file. The confound time series derived from head motion estimates and global signals were expanded with the inclusion of temporal derivatives and quadratic terms for each[106]. Frames that exceeded a threshold of 0.5 mm FD or 1.5 standardized DVARS were annotated as motion outliers. All resamplings can be performed with *a single interpolation step* by composing all the pertinent transformations (i.e., head-motion transform matrices, susceptibility distortion correction when available, and co-registrations to anatomical and output spaces). Gridded (volumetric) resamplings were performed using antsApply-Transforms (ANTs), configured with Lanczos interpolation to minimize the smoothing effects of other kernels[107]. Non-gridded (surface) resamplings were performed using mri_vol2surf (FreeSurfer).

Many internal operations of *fMRIPrep* use *Nilearn* 0.6.2 (ref. 108, RRID:SCR_001362), mostly within the functional processing workflow. For more details of the pipeline, see the section corresponding to workflows in *fMRIPrep*'s documentation (https://fmriprep.org/en/latest/workflows.html).

### MRI data quality checks

We assessed fMRI data quality based on measures of head motion as a potential source for noise and artifacts. We investigated framewise displacement for each run and participant and marked each volume as an outlier if it exceeded a threshold of 0.5 mm (criterion used by fMRIPrep). On average, motion was relatively low (mean framewise displacement across participants: $M = 0.15$ mm, $SD = 0.04$ mm, range $= 0.07$–0.25 mm; mean percentage of outlier volumes: $M = 0.91\%$, $SD = 1.34\%$, range $= 0$–6.09%; all participants below our criterion of max. 10% outlier volumes for inclusion in the main data analyses). To control for head motion, we included motion parameters as confounds in first-level GLMs (see below).

Additionally, we assessed MRI data quality using the tool MRIQC (version 0.16.1) which calculates a set of image quality metrics for both functional and anatomical image data.

### Region of interest (ROI) definition

For our hypothesis of a grid-like representation in the entorhinal cortex, we used participant-specific bilateral entorhinal cortex masks created by FreeSurfer segmentations of the participants' anatomical images during preprocessing with fMRIPrep (FreeSurfer labels 1006 & 2006, $M = 269$ voxels, $SD = 42$ voxels). For small volume correction within the entorhinal cortex on the group level, we combined both participant-specific anatomy and MNI standard atlas labeling. For this purpose, we first transformed the participant-specific masks to MNI standard space and created the union of all masks across participants. We then intersected this union mask with the entorhinal cortex mask of the Juelich Histological Atlas provided by FSL and thresholded at 50% probability. Finally, we intersected this mask with the whole-brain group mask comprising only voxels shared across participants (resulting mask used for small volume correction: 411 voxels; Supplementary Fig. 4a). To further explore vmPFC representations, we defined two ROIs as spheres with a 7 mm radius (1) around the peak voxel of our value difference analysis in vmPFC (MNI peak voxel

coordinates: 3,42,−8; 89 voxels) and (2) around the peak voxel of the hexadirectional effect reported by Constantinescu et al.[39] in vmPFC (MNI peak voxel coordinates: 16,54,−2; 95 voxels).

For our choice decoding hypothesis, we leveraged neural responses to category-specific stimuli (faces, tools, scenes, body parts) in category-selective regions of the occipital-temporal cortex. We created participant-specific occipital-temporal ROI masks as follows. First, we thresholded occipital and temporal lobe probability masks of the MNI Structural Atlas provided by FSL (version 6.0.3) at a threshold of 25% and created their union. We then transformed this MNI-based mask to each participants' native space using ANTS (version 2.3.5) and resampled it to the resolution of the functional data based on transformation files created during preprocessing with fMRIPrep. We intersected these with participant-specific gray matter masks. For this purpose, we thresholded gray matter probability masks created by fMRIPrep's segmentation of the anatomical image at a threshold of 50% and resampled them to the functional resolution. In the decoding analysis, we used these participant-specific gray matter occipital-temporal masks for additional feature selection based on univariate stimulus-category effects in the PVT training data (see below). The final masks used for choice decoding comprised 2235 voxels on average ($SD = 173$ voxels; Supplementary Fig. 8a).

### General set-up of first level general linear models (GLMs)

For our fMRI data analyses, we used both univariate and multivariate approaches. For both approaches, we modeled the fMRI data using event-related GLMs. In the following, we briefly describe commonalities of GLMs across analyses.

We implemented run-wise first level GLMs using the First-LevelModel class of the nilearn package. GLMs were computed within a brain mask (either in participants' native space or in MNI standard space, stated for each analysis below). To create a common brain mask for all runs, we resampled the anatomical brain mask in native or MNI space created during preprocessing with fMRIPrep to the resolution of the functional data. Task-related regressors in the GLMs were convolved with the Glover haemodynamic response function (HRF). Temporal autocorrelation in the fMRI data was accounted for using an autoregressive AR(1) model. For univariate analyses, the data were spatially smoothed with a 6 mm full-width at half maximum Gaussian filter (FWHM). For the multivariate choice decoding analysis, no smoothing was applied to preserve differences between voxels.

All GLMs included the following regressors for task-related events of no interest: two regressors for left and right button presses with a stick duration as well as a regressor modeling the end-of-block notification screen at the end of a run. To control for noise signals in the fMRI data, the GLMs included 37 confound regressors estimated during preprocessing with fMRIPrep. Following the denoising strategy proposed by Satterthwaite et al.[106], these confounds included 24 motion parameters (6 basic translation / rotation parameters, 6 temporal derivatives of these and 12 quadratic terms of the basic parameters and their derivatives) as well as 12 global signal parameters (3 basic average CSF, WM and global signal parameters, 3 temporal derivatives of these and 6 quadratic terms of the basic parameters and their derivatives). Additionally, the confounds included framewise displacement as a summary metric of frame-to-frame head motion. Furthermore, the GLMs included discrete cosine-basis regressors estimated by fMRIPrep to account for temporal low-frequency signal drifts.

### Analysis of hexadirectional signals (grid-like representation)

To investigate whether the entorhinal cortex encodes the abstract value space using a grid-like representation, we implemented the hexadirectional analysis approach by Doeller et al.[52]. Grid cells in the entorhinal cortex are characterized by their regular hexagonal firing pattern which translates to hexadirectional activity modulations

during navigation in fMRI, with higher activity for navigation in directions aligned with the putative grid orientation (phase of the hexadirectional signal) than for trajectories misaligned with the putative grid orientation. The analysis consists of two steps: In the first step, the grid orientation is estimated and in the second step the prediction of hexadirectional modulation according to the grid orientation is tested using independent data. Here, we tested for such a hexadirectional modulation as a function of trajectories through our value space. We implemented a cross-validation procedure, estimating the putative entorhinal grid orientation using three of four task runs and testing for a hexadirectional modulation aligned to the orientation in the left-out test run[52,53].

We implemented this cross-validation procedure on fMRI data in participants' native space to enable estimations of grid orientations in participant-specific entorhinal cortex ROIs.

In the estimation set (three of four runs, GLM1), the GLM for each run included a main effect regressor modeling trajectories including all time points and a main effect regressor modeling feedback periods. The regressors were modeled with the actual onset and durations of the events during the task. The trajectory regressor was accompanied by two parametrically modulated regressors. These modulations reflected the sine and cosine of the direction (angle) θ of the trajectory with 60° (6-fold) periodicity ($\sin(6 * \theta_t)$ and $\cos(6 * \theta_t)$). Values for both regressors were demeaned. Effect sizes of the regressors were averaged across runs of the estimation set (fixed effects). We then used the effect sizes of the sine ($\beta_{\sin}$) and cosine ($\beta_{\cos}$) regressors to estimate the grid orientation in 60°-space (range [0,60°]) in each voxel of the entorhinal cortex as follows:

$$\Theta = \frac{\arctan\left(\frac{\beta_{\sin}}{\beta_{\cos}}\right)}{6}$$

Subsequently, we calculated the mean orientation across voxels of the entorhinal cortex with a weighting of the voxels by their amplitude of the hexadirectional modulation ($\sqrt{\beta_{\sin}^2 + \beta_{\cos}^2}$)[109]. For this purpose, we first transformed voxel orientations back to 360°-space to allow for calculations of trigonometric functions (multiplication by 6). We then transformed these orientations and the amplitudes from polar to cartesian coordinates and took the mean separately for both dimensions. Afterwards, we transformed the mean back to polar coordinates and subsequently transformed the mean orientation back to 60°-space.

In the independent test set (left-out run, GLM2), the GLM included a main effect regressor modeling trajectories including all time points and a main effect regressor modeling feedback periods. The regressors were modeled with the actual onset and durations of the events during the task. The trajectory regressor was accompanied by a parametrically modulated regressor reflecting a six-fold (hexadirectional) sinusoidal modulation based on the estimated mean entorhinal grid orientation ($\cos(6 * (\theta_t - \Theta))$). Values for the regressor were demeaned. Effect sizes of the parametric cosine regressor were averaged across the four cross-validation folds (fixed effects) to obtain an overall effect size.

For group level statistics, we first transformed effect size images of the parametric cosine regressor to MNI standard space. We then performed an analysis with small volume correction based on our a priori ROI of the entorhinal cortex (see ROI definition). Additionally, we performed a whole-brain analysis based on a whole-brain group mask comprising only voxels shared across participants. We tested significance across participants using non-parametric permutation testing implemented in FSL Randomize with 10000 permutations. We used one-sided tests as the predicted direction of the hexadirectional effect is inherently positive (higher activity for navigation in directions aligned vs. misaligned with the grid orientation). We used threshold-free cluster enhancement and corrected for multiple comparisons with

family-wise error rate ($p_{FWE} < 0.05$) within the small volume correction mask and whole-brain. For exploration of whole-brain effects at an uncorrected threshold of $p < 0.001$, we extracted cluster information using nilearn and respective brain region labels of the Harvard-Oxford Cortical Structural Atlas, Harvard-Oxford Subcortical Structural Atlas and Juelich Histological Atlas using FSL atlasquery.

To visualize the hexadirectional effect in the significant entorhinal cluster, we implemented an additional GLM for the test set (left-out run, GLM2) by binning trajectories based on directions. To this end, we sorted trajectories into bins of 30° based on the mean entorhinal grid orientation (+/− 15° of the grid orientation and multiples of 60°). This resulted in 12 trajectory bin regressors, 6 reflecting trajectories aligned and 6 misaligned with the grid orientation. In this GLM, we therefore modeled trajectories using the 12 bin regressors and a main effect regressor for all trajectories capturing the mean. Effect size images were averaged across the four cross-validation folds (fixed effects) and transformed to MNI standard space. We extracted the mean effect size of each trajectory bin in the significant cluster.

In control analyses, we investigated the relationship of the hexadirectional effect with the spatial and temporal stability of voxel-wise grid orientations in the significant entorhinal cluster. Spatial stability refers to similarity of orientations across voxels within the significant cluster. To investigate spatial stability, we first transformed effect size images of the sine ($\beta_{\sin}$) and cosine ($\beta_{\cos}$) regressors of the estimation GLM (GLM1) based on all runs (to increase power) to MNI standard space. We then estimated voxel orientations as described above. For each participant, we tested deviation from a uniform distribution of voxel orientations in the significant cluster using a Rayleigh test for non-uniformity of circular data (implemented in the package pycircstat, version 0.0.2, https://github.com/circstat/pycircstat). Across participants, we calculated a Pearson correlation between the Rayleigh z-statistic and the hexadirectional effect. To control for similarity of voxels introduced by smoothing, we additionally investigated spatial stability using unsmoothed data. Temporal stability refers to similarity of orientations within a voxel across time. To investigate temporal stability, we followed the logic of the cross-validation procedure described above and additionally estimated orientations in the left-out test run. For each voxel and for each cross-validation fold, we calculated the orientation difference between the estimation and the test set. Subsequently, we averaged orientation differences across folds and classified voxels as stable if their mean orientation difference was within 15°. Across participants, we tested whether the percentage of stable voxels was different from 50% using a one-sample t-test. Furthermore, we calculated a Pearson correlation between the percentage of stable voxels and the hexadirectional effect.

In addition to the small volume correction analysis, we conducted a complementary ROI analysis based on participants' individual entorhinal Freesurfer masks. In this ROI analysis, we also investigated the specificity of a hexadirectional (6-fold) modulation of activity in line with grid cell firing by performing control analyses for a four-, five-, seven- and eight-fold modulation (same cross-validation procedure as described above). On the group level, we tested whether effect sizes were different from 0 using one-sample t-tests (with α = 0.01, Bonferroni-corrected for five tests).

We assessed the relationship between the hexadirectional effect in the significant entorhinal cluster and overall task performance using a Pearson correlation.

Lastly, we performed exploratory analyses to investigate the relationship between the entorhinal grid system and the underlying value space. First, we investigated clustering of orientations in the significant entorhinal cluster. To this end, we estimated each participant's mean orientation in the significant cluster as described above (based on all runs to increase power). We tested whether orientations across participants cluster around 45° using a V-Test (implemented in the package astropy[110], version 5.0). Secondly, we tested whether the

magnitude of the hexadirectional modulation differed between high- and low-value-areas of the value space. For this purpose, we performed a median split of trajectories according to their mean value. This meant contrasting trajectories in the lower left triangle of the space (low-value-area) with trajectories in the upper right triangle of the space (high-value-area). We note that this median split led to a substantial reduction of available trajectories per value condition and an unbalanced sampling of directions between the conditions, rendering this analysis less robust. To examine sampling of directions, we counted the frequency of directions per condition per participant and tested for differences using a repeated measures ANOVA across participants with the factors direction and value condition. We repeated the cross-validated hexadirectional analysis described above, with two changes: First, we based this analysis on the significant entorhinal cluster, both for the estimation of the grid orientation (GLM1) and for testing the hexadirectional effect on the group level (ROI analysis). Note that this analysis is still unbiased as we were interested in the difference of the hexadirectional effect between value conditions. Secondly, the GLMs for the estimation and test set estimated effects separately for the value conditions. More specifically, the estimation set (GLM1) included separate main effect and sine- and cosine-parametrically modulated regressors for each value condition and the grid orientation was estimated separately for each value condition. Analogously, the independent test set (left-out run, GLM2) included separate main effect and cosine-parametrically modulated regressors for each value condition. We averaged effect sizes across voxels of the ROI (significant cluster of the overall hexadirectional effect). Across participants, we tested for a difference between value conditions using a related-samples $t$-test as well as for individual effects using one-sample $t$-tests (one-sided).

To further explore vmPFC representations, we repeated the cross-validated hexadirectional analysis while estimating the putative grid orientation in vmPFC (two vmPFC ROIs, see Region of interest (ROI) definition, analysis directly in MNI space). For each ROI, we then tested whether effect sizes were different from 0 using one-sample $t$-tests.

## Value difference analysis

To investigate whether fMRI activity is modulated by the value difference between options during choices, we implemented a GLM with three main effect regressors: one regressor modeled the observation phase (initial time points) of the trajectories, one regressor modeled choice time points and one regressor modeled feedback periods. The regressors were modeled with the actual onset and durations of the events during the task. The choice time point regressor was accompanied by two parametrically modulated regressors. These modulations reflected the value of the chosen option and the value of the unchosen option, as estimated by the prospective Rescorla-Wagner model. Values for both regressors were demeaned so that they were orthogonal to the main effect regressor. We then contrasted the estimated effect sizes of the chosen value vs. the unchosen value regressor [1, −1] to test for a modulation of activity by the value difference. Contrasts were averaged across runs (fixed effects).

To investigate whether fMRI activity is modulated specifically by the prospective component of the value difference, we changed the two parametrically modulated regressors for the choice time points as follows: One regressor reflected the difference in the value estimate of the chosen option between the prospective Rescorla-Wagner model and the original Rescorla-Wagner model (non-prospective). Analogously, the other regressor reflected the difference in the value estimate of the unchosen option between the prospective Rescorla-Wagner model and the original Rescorla–Wagner model (non-prospective). We excluded one participant from this analysis because the value estimates of the two models were very similar (mean difference = 0.16, participant with lowest learning rate in the prospective Rescorla–Wagner model).

In three control analyses, (1) we added an additional parametrically modulated regressor for choice time points reflecting reaction time, (2) we restricted the parametrically modulated value regressors to correct trials only and (3) we added an additional parametrically modulated regressor for choice time points reflecting the distance between the choice location and the 45°-diagonal. Reaction times were log-transformed and demeaned, the distance between the choice location and the diagonal was demeaned.

We computed these GLMs on fMRI data in MNI standard space. For group level statistics, we performed whole-brain analyses based on a whole-brain group mask comprising only voxels shared across participants. We tested the significance of contrasts across participants using non-parametric permutation testing implemented in FSL Randomize with 10000 permutations. We used threshold-free cluster enhancement and corrected for multiple comparisons with family-wise error rate ($p_{FWE} < 0.05$). We extracted cluster information using nilearn and respective brain region labels of the Harvard–Oxford Cortical Structural Atlas, Harvard–Oxford Subcortical Structural Atlas and Juelich Histological Atlas using FSL atlasquery.

Furthermore, we visualized the value difference effect in the significant vmPFC cluster by showing time courses of the effects of the chosen and unchosen value time-locked to the onset of the choice time points. For this purpose, we extracted the preprocessed fMRI time series of voxels in the vmPFC cluster. Analogously to general first-level modeling, we spatially smoothed (6 mm FWHM) and cleaned the data by regressing out confounds and temporal low-frequency signal drifts. We z-scored each voxel's time series, averaged them across voxels of the cluster and interpolated the signal (cubic spline interpolation). For each choice time point, we extracted the cluster signal in a time window of 16 s, time-locked to 1 s before onset of the choice time point in steps of 0.1 s until 15 s after onset. Subsequently, we ran a linear regression across choice time points of a run for each time step (in steps of 0.1 s), with the regressors chosen value, unchosen value, trial number and an intercept. Value and trial regressors were demeaned beforehand. We extracted effect sizes of the chosen and unchosen value regressor for all time steps and averaged them across runs for a given participant. Lastly, we averaged these time courses across participants for visualization.

We tested for a relationship between the value difference effect in the vmPFC cluster and task performance using a Pearson correlation.

## Choice decoding analysis (Supplementary Figs. 8–10)

To investigate whether the occipital-temporal cortex represents the high-value option more strongly than the low-value option during choices, we implemented the following decoding analysis. Using independent data from the picture viewing task (PVT) which took place before the prospective decision making task, we trained a decoder (support vector classifier) on occipital-temporal cortex voxels to distinguish neural activation patterns of the four category-specific stimuli (faces, tools, scenes, body parts). We then applied this decoder to neural activation patterns of choice time points in the prospective decision making task. We performed this analysis in participants' native space.

To estimate neural activation patterns of stimuli in the PVT training data, we implemented a Least-Squares Separate GLM approach. More specifically, we ran 57 single-trial-GLMs, one for each regular trial of the task. Each GLM included one regressor modeling the trial of interest and one regressor modeling all other regular trials. Test trials were modeled in a separate regressor. The regressors were modeled with the actual onset and durations of the events during the task. We used z-scores of the trial regressors for the next steps (56 z-scores, the first trial was discarded to allow for balanced sampling of stimulus categories: 14 trials per category).

Based on the PVT training data, we created the final participant-specific ROI masks used for the decoding analysis. We combined the

predefined anatomical gray matter occipital-temporal masks with the functional PVT data to select category-stimuli-responsive voxels (features). To this end, we extracted trial-wise z-scores for each voxel within the predefined anatomical mask. We z-standardized them across trials and performed univariate feature selection by computing ANOVA F-values between each feature and the trial labels. We selected those 20% of the voxels with the highest F-values. The resulting masks were used for the decoding analysis in the next steps.

As a control, we first examined how well we could decode stimulus category within the PVT, before applying the decoder to the decision making task. For this purpose, we extracted trial-wise z-scores for each voxel within the decoding ROI mask. We implemented a 7-fold cross-validation scheme with 8 left-out test trials (2 trials per category) and 48 training trials. We trained a decoder to distinguish neural activation patterns of the four category-specific stimuli and to predict labels for the left-out test trials (support vector classifier, regularization parameter C = 1.0, kernel = rbf). We z-standardized trial-wise z-scores within the training set and applied the standardization parameters to the test trials. We assessed accuracy as the proportion of correctly predicted trials, averaged over cross-validation folds. Note that we conducted this analysis using the decoding ROI mask based on previous univariate feature selection across all trials of the PVT (see above). We did so as this was the mask used for the following across-task decoding (PVT to prospective decision making task) and the stimulus category decoding within the PVT served only as a control.

Next, we aimed to investigate stimulus representations during choices in the prospective decision making task. To first estimate neural activation patterns of choice time points in the prospective decision making task, we implemented the following GLM. One regressor modeled the observation phase (initial time points) of the trajectories and one regressor modeled feedback periods. Each choice time point (18 choice time points per run) was modeled in a separate regressor. The regressors were modeled with the actual onset and durations of the events during the task. We used z-scores of the choice time point regressors as test data for the across-task decoding analysis.

For each voxel within the decoding ROI mask, we extracted trial-wise z-scores of the PVT as training data and choice z-scores of the prospective decision making task as test data. We z-standardized the data run-wise. We then trained a decoder to distinguish neural activation patterns of the four category-specific stimuli based on the PVT data (support vector classifier, regularization parameter C = 1.0, kernel = rbf, probability = True to enable probability estimates). Subsequently, we applied this decoder to the neural activation patterns of choices in the prospective decision making task. More specifically, we extracted the probabilities which the decoder assigned to each of the four stimuli and computed two difference scores for each choice. First, we compared the probabilities assigned to the two stimuli presented on-screen during choice: probability of the stimulus with the objectively higher value vs. probability of the stimulus with the objectively lower value. Secondly, we compared the probabilities assigned to the two value-congruent stimuli which were not presented on-screen during choice (but during the time point before): probability of the congruent high-value stimulus vs. the congruent low-value stimulus. To compare these difference scores against chance level performance of the decoder, we implemented a permutation test, repeating this procedure 1000 times with randomly permuted trial labels in the PVT training data. For each choice, we then converted the original difference scores to z-scores based on the null distribution generated by the permutations. Lastly, we averaged z-scores across choices to obtain two summary scores per participant. On the group level, we tested participant-specific z-scores against 0 using one-sample t-tests. Furthermore, we calculated Pearson correlations between the z-scores and task performance.

When comparing on-screen and congruent off-screen stimuli separately, the temporal proximity of their presentations during time points within a trajectory might render disentangling their effects difficult. To control for the temporal proximity to some extent, we repeated the analysis using only those choices which sampled the switch time point as a control. In this case, the direction of the effect during choice (high-value vs. low-value, especially for the comparison of the congruent stimuli) should be different from the direction of the effect at the time point before the switch (pre).

Given that participants performed the task very well (M = 87.70%), the stronger representation of the high-value option compared to the low-value option might be driven by selective attention towards the chosen option. To disentangle a value from an attention/choice effect, we repeated the analysis using only incorrectly answered trials (note the very low number of available incorrect trials). In this case, a stronger representation of the high vs. low value option would suggest a value effect while the opposite pattern, a stronger representation of the low-value option, would suggest an attention / choice effect.

### Reporting summary

Further information on research design is available in the Nature Portfolio Reporting Summary linked to this article.

## Data availability

Data to reproduce the statistical analyses reported in this paper are available on the Open Science Framework (https://osf.io/z4k5v/). Task stimuli are available in public stimulus datasets[77–82]. The Harvard-Oxford Cortical and Subcortical Structural Atlases and the Juelich Histological Atlas used for the neuroimaging analyses are provided by FSL. Source data are provided with this paper.

## Code availability

Analysis code is available on Github (https://github.com/nitschalex/Paper_Value_Space).

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

## Acknowledgements

We would like to thank Theo Schaefer and Felix Deilmann for joint efforts in setting up MRI-compatibility of Psychopy tasks and co-development of scripts implementing MRI data analyses in Python. We would also like to thank Ulrike Horn for providing a Python script to interact with the MRI button box. We would like to thank Kerstin Schumer, Max Schulz, Anke Kummer, Simone Wipper, Sylvie Neubert, Mandy Jochemko, Nicole Pampus and Manuela Hofmann for help with data collection. Further-more, we would like to thank Arthur Levasseur for help with piloting a previous task version. This study was further developed following pre-vious work on a value space by Naomi de Haas, whom we would like to thank for her previous work as well as helpful input and discussions. We would also like to thank past members of the Doellerlab, including Stephanie Theves, for previous discussions regarding abstract spaces. We thank the University of Minnesota Center for Magnetic Resonance Research for the provision of the multiband EPI sequence software and Toralf Mildner, Joeran Lepsien and colleagues of the Doellerlab as well as the MPI research group Adaptive Memory for joint efforts in further MRI sequence piloting. We would also like to thank all colleagues of the Doellerlab for helpful discussions of the study. This work was supported by the Max Planck Society. N.W.S. is funded by an Independent Max Planck Research Group grant awarded by the Max Planck Society (M.TN.A.BILD0004) and a Starting Grant from the European Union (ERC-2019-StG REPLAY-852669) and the Federal Ministry of Education and Research (BMBF) and the Free and Hanseatic City of Hamburg under the Excellence Strategy of the Federal Government and the Länder. C.F.D. is supported by the Max Planck Society, the European Research Council (ERC-CoG GEOCOG 724836), the Kavli Foundation, the Jebsen Foundation, Helse Midt Norge and The Research Council of Norway (223262/F50, 197467/F50).

## Author contributions

A.N., N.W.S. and C.F.D. conceived the experiment. A.N., M.M.G., N.W.S. and C.F.D. designed the experiment and developed the tasks. A.N. acquired the data. A.N., M.M.G., J.L.S.B., N.W.S., and C.F.D. planned the analyses. A.N. performed the data analyses and A.N., M.M.G., J.L.S.B., N.W.S., and C.F.D. discussed the results. A.N. wrote the manuscript with input from M.M.G., J.L.S.B., N.W.S., and C.F.D.

## Funding

## Competing interests

The authors declare no competing interests.
