## [Peer Review File · Nature Communications]

Grid-like entorhinal representation of an abstract value space
during prospective decision makingREVIEWER COMMENTS

Reviewer #1 (Remarks to the Author):

In this study, the authors build upon results in monkeys that suggest that value information might be organized in the brain via map-like representations. The authors had subjects perform a decision-making task in which the reward contingencies changed by moving along a trajectory in a two-dimensional value space. They then scanned the subjects and used a sophisticated statistical pipeline to determine whether neural activity in entorhinal cortex was consistent with a grid-like representation. The lab has developed this approach over several years and these methods have been well validated. They revealed that indeed value information was encoded in a grid-like way supporting the idea that value is represented as part of a cognitive map.

Overall, the paper is of very high quality. The behavioral task has been devised in a careful and thoughtful way that includes all the experimental conditions and controls that I would want to see. The analytical methods are sophisticated, rigorous, and appropriate. The results are clearly presented and support the author's conclusions. My comments are largely minor in nature.

1. I think it is interesting that the grid aligned to the 45° diagonal, and I agree with the authors' interpretation of this effect. They argue that this line is important since any trajectory that was parallel to 45° (and presumably 225°) cannot involve a switch in value. I wish they had done more to support this claim behaviorally. I found myself squinting at Supplementary Figure 2D to see where performance was indeed higher for the $45^\circ/225^\circ$ trajectories. Equally, one might also anticipate higher performance at $135^\circ/315^\circ$ trajectories since those would be perpendicular to the 45° line and therefore easier to judge whether they would intersect. I think the authors might want to do a more formal analysis of subjects' behavior to see whether this is true. It would also be interesting to see where the subjects who show larger effects of trajectory direction on behavioral performance have stronger encoding of the grid-like code.

2. The section describing how visual areas encoded the high value option more strongly was

probably the weakest part of the paper since the interpretation of these results is confounded by the fact that the subjects almost certainly attended to the high value option more than the low value option. The section felt out of place and detracted from the focus of the paper. I would recommend scrapping it, or at least moving it into the supplementary.

3. I think it is quite surprising that there was no grid code in vmPFC especially given results from folks like the Rushworth lab, Behrens lab, and Boorman lab. I think it would be useful to devote at least some of this discussion to consideration of this inconsistency with the prior literature.

4. Figure 1B: I'm not sure what the "arrow in petrol" means. Is "petrol" supposed to be a color? I've never heard of it. I think "light blue" would probably be a better descriptor.

Reviewer #2 (Remarks to the Author):

The manuscript by Nitsch and colleagues reports a human fMRI study that measured neural representations of abstract value spaces during a decision making task. In the experiment, subjects were shown images corresponding to two different latent categories (A and B) along with values associated with each category. The critical feature of the task is that the values associated with these latent categories changed across trials in a consistent way, such that the changes could be described as a trajectory within the 2-d value space. The main question addressed in the study is whether this movement in an unseen 2-d value space was associated with grid-like responses (hexagonal coding) in the entorhinal cortex. While grid-like responses have been observed in entorhinal cortex for physical spaces in humans and rodents, and several fMRI studies have found grid-like coding for more conceptual spaces, the specific question of whether grid-like coding is observed for value spaces in human entorhinal cortex is novel. Indeed, the paper demonstrates grid-like effects within entorhinal cortex. These effects were selective to entorhinal cortex and also selective to a hexagonal coding scheme. Additional analyses show that value differences between categories (a 1 d representation) are reflected by widespread fMRI activations and that visual categories associated with higher rewards are preferentially represented in occipitotemporal cortex.

The paper is very well written, clearly organized, and well motivated from the prior literature. The key results are robust and there are a number of converging analyses to reinforce the main ideas. Thus, it is overall a very solid paper. The only notable weakness, in my view, is that the advance relative to prior studies is somewhat modest. There have now been several demonstrations of grid-like responses for non-physical spaces (in humans and rodents), so the extension to value-based decision making is not necessarily a major advance. Likewise, some of the secondary analyses are useful, but I think the framing of the results makes them sound a bit more interesting than they actually are. Thus, my assessment is that this is a very nice and rigorous paper that adds to an exciting literature (grid-like coding for conceptual information) but the degree of conceptual advance is perhaps borderline with respect to the standards for this journal.

COMMENTS

1. The main findings of grid-like coding in entorhinal cortex are, essentially, an extension of prior findings of similar effects for other conceptual spaces. I still find this striking and interesting, but the main goal of the current study appears to be to test whether this form of coding extends to value spaces. It is not clear that there is a meaningful alternative or that the paper tests competing ideas. Thus, it functions more of a nice demonstration / extension.

2. The stronger representation of higher value stimuli in occipitotemporal cortex makes sense, but is also not particularly exciting. This almost feels like a validation that subjects paid more attention to the stimuli that they were selecting. In other words, rather than framing this in terms of OTC 'caring' about value, it could simply be that OTC encoded what was attended/selected. Given that higher value items tended to be chosen much more often than lower value items, there is no simple way to disentangle value from choice. I do think the effects for value congruent stimuli (stimuli not actually on the screen) are neat, but again, I do not see how to tease apart a value representation, per se, from selective attention.

3. Likewise, the prospective value effects (1d space) are worth reporting, but not particularly exciting because it is very hard to disentangle a representation of value, per se, from the decision difficulty. I do appreciate that the authors controlled for RT, but it is hard to fully control for difficulty (e.g., the authors used log-transformed RT, which makes an assumption about how difficulty scales with RT). The fact that the 1d value effects were so widespread in the brain also suggests that something like effort or difficulty is a reasonable interpretation. It is not clear that there is any way, in the current design, to completely tease apart the value estimate from other things that are bound to be confounded with value estimates.

4. I am wondering if subjects tended to settle into a strategy of explicitly “doing the math” to make their decisions. For example, after the 2nd time point, subjects know the increment and direction of change (e.g., +5 for variable A and -9 for variable B). Did subjects report tracking this explicitly? At least to me, this seems like it would be the best way to perform the task. Obviously, this strategy would not involve an integration into a 2d space. As such, I don’t see how this strategy would explain the fMRI results. So, my question is not necessarily a concern; rather, I am just curious what strategy subjects tended to use and whether this might constrain interpretations at all? The Discussion says that subjects did not report using a 2D (integrated) space, but I did not see any mention of strategies that subjects DID report using.

MINOR

5. p. 8: “We found that the likelihood of correct choices increased with increasing distance of the choice location to the diagonal (Fig. 2c-d; $t(44) = 8.03$, $p < .001$).” Should this instead say “distance of the choice location FROM the diagonal”? As written, it sounds like accuracy increased as choices got closer to the diagonal, but I assume the authors mean the opposite?

6. Fig. 3 legend: panel “f” is referred to as panel “e”

Response to the reviewers' comments

1 Reviewer 1

1.1 General remarks

In this study, the authors build upon results in monkeys that suggest that value information might be organized in the brain via map-like representations. The authors had subjects perform a decision-making task in which the reward contingencies changed by moving along a trajectory in a two-dimensional value space. They then scanned the subjects and used a sophisticated statistical pipeline to determine whether neural activity in entorhinal cortex was consistent with a grid-like representation. The lab has developed this approach over several years and these methods have been well validated. They revealed that indeed value information was encoded in a grid-like way supporting the idea that value is represented as part of a cognitive map.

Overall, the paper is of very high quality. The behavioral task has been devised in a careful and thoughtful way that includes all the experimental conditions and controls that I would want to see. The analytical methods are sophisticated, rigorous, and appropriate. The results are clearly presented and support the author's conclusions. My comments are largely minor in nature.

We thank the reviewer for their very positive evaluation of our manuscript and their thoughtful comments. By addressing their comments, we have – amongst other aspects – included further behavioral support for our interpretation of the alignment of the grid system to 45° and an additional discussion point regarding the absence of evidence for a grid-like representation in vmPFC. We believe that these suggested changes have further strengthened our manuscript. Please find our detailed point-by-point responses below.

1.2 Relevance of the 45° reference direction

I think it is interesting that the grid aligned to the 45° diagonal, and I agree with the authors' interpretation of this effect. They argue that this line is important since any trajectory that was parallel to 45° (and presumably 225°) cannot involve a switch in value. I wish they had done more to support this claim behaviorally. I found myself squinting at Supplementary Figure 2D to see where performance was indeed higher for the 45°/225° trajectories. Equally, one might also anticipate higher performance at 135°/315° trajectories since those would be perpendicular to the 45° line and therefore easier to judge whether they would intersect. I think the authors might want to do a more formal analysis of subjects' behavior to see whether this is true. It would also be interesting to see where the subjects who show larger effects of trajectory direction on behavioral performance have stronger encoding of the grid-like code.

We thank the reviewer for this suggestion. Indeed, it would be interesting to see whether the relevance of directions parallel to the 45°-diagonal is also reflected in behavior. To address

this question, we grouped trajectories according to directions approximately parallel to the 45°-diagonal (sampled directions: 40°, 50°, 220°, 230°), directions approximately perpendicular to the 45°-diagonal (sampled directions: 130°, 140°, 310°, 320°) and all other directions. For this comparison, we note that parallel trajectories inherently involved fewer switches of the more valuable option than perpendicular trajectories (e.g., parallel trajectories involved more often the same objective value for both options at the choice time point). We thus tested whether directions influenced performance by taking into account possible differences due to the differences in switches using a repeated measures ANOVA with the within-subject factors direction and switch vs. non-switch trajectory (two participants excluded due to missing data for some conditions; Review Figure 1a). We observed significant main effects of direction ($F(2,86) = 3.20, p = .045$) and switch ($F(1,43) = 59.06, p < .001$) as well as a significant interaction between direction and switch ($F(2,86) = 7.18, p = .001$). We further investigated the interaction effect using post-hoc pairwise tests with Bonferroni correction ($\alpha = .008$, Bonferroni-corrected for 6 pairwise tests). This revealed significantly higher performance for parallel vs. perpendicular directions in switch trajectories only ($t(43) = 2.90, p = .005$; comparison parallel vs. other in switch trajectories: $t(43) = 2.67, p = .014$ n.s. after correction; all other pairwise comparisons n.s. $p > .008$).

Next, we investigated whether the performance benefit for parallel vs. perpendicular directions in switch trajectories was related to the grid-like representation. We did not observe any significant correlation, neither with the magnitude of hexadirectional modulation (Review Figure 1b; Spearman $r(42) = .03, p = .82$) nor with the difference between the estimated grid orientation and 45° (Review Figure 1c; Spearman $r(42) = -.05, p = .73$; also no significant correlations when tested with the performance benefit for parallel vs. other directions).

Review Figure 1 | Relevance of the 45° reference direction. **a** Performance for trajectories with directions approximately parallel to the 45°-diagonal, approximately perpendicular to the 45°-diagonal and all other directions, separately for switch and non-switch trajectories. There was a significant interaction between switch vs. non-switch trajectories and direction, with post-hoc tests indicating significantly higher performance for parallel vs. perpendicular directions in switch trajectories. **b** There was no significant correlation between the magnitude of hexadirectional modulation and the performance benefit for parallel vs. perpendicular directions in switch trajectories. **c** There was no significant correlation between the difference between the estimated grid orientation and 45° and the performance benefit for parallel vs. perpendicular directions in switch trajectories. * $p < .05$ corr.

As suggested by the reviewer, we think that the performance benefit for parallel directions in switch trajectories supports our interpretation of the alignment of the grid system to 45° as the most informative axis in our value space. We now describe this new aspect in the Results, Discussion and Methods sections and include it in Supplementary Fig. 2f.

Results, page 10:

In line with this idea, participants' performance was higher for trajectories with directions approximately parallel to the 45°-diagonal in switch trajectories (Supplementary Fig. 2f; interaction effect direction and switch: $F(2,86) = 7.18$, $p = .001$; post-hoc test parallel vs. perpendicular in switch trajectories: $t(43) = 2.90$, $p = .005$, with $\alpha = .008$, Bonferroni-corrected for 6 pairwise tests; all other pairwise comparisons n.s. $p > .008$).

Discussion, page 14:

More specifically, our results suggested an anchoring of grid orientations around 45° and participants' performance was also increased for directions parallel to 45° in switch trajectories.

Methods, page 27:

Furthermore, we compared performance for directions approximately parallel to the 45°-diagonal (sampled directions: 40°, 50°, 220°, 230°), directions approximately perpendicular to the 45°-diagonal (sampled directions: 130°, 140°, 310°, 320°) and all other directions, by taking into account possible differences due to the differences in switches, using a repeated measures ANOVA with the within-subject factors direction and switch vs. non-switch trajectory (two participants excluded due to missing data for some conditions). We further investigated the interaction effect using post-hoc pairwise related-samples t-tests with Bonferroni correction ($\alpha = .008$, Bonferroni-corrected for 6 pairwise tests).

1.3 Choice decoding analysis

The section describing how visual areas encoded the high value option more strongly was probably the weakest part of the paper since the interpretation of these results is confounded by the fact that the subjects almost certainly attended to the high value option more than the low value option. The section felt out of place and detracted from the focus of the paper. I would recommend scrapping it, or at least moving it into the supplementary.

We thank the reviewer for bringing up this point (see also below for a similar point made by Reviewer 2, point 2.3). Given that participants performed the task very well ($M = 87.70\%$), we agree that the stronger representation of the high-value option compared to the low-value option might also reflect selective attention towards the chosen option.

In our view, the only possibility to disentangle a value from an attention / choice effect is to conduct the decoding analysis using only incorrectly answered trials. In this case, a stronger representation of the high vs. low value option would suggest a value effect while the opposite pattern, a stronger representation of the low-value option, would suggest an attention / choice effect. We conducted this control analysis and despite the very low number of available incorrect trials, we observed significantly lower probabilities for the high-value vs. the low-value stimulus, both when comparing on-screen stimuli (Review Figure 2a-b; $t(45) = -5.57$, $p < .001$) as well as when comparing congruent off-screen stimuli (Review Figure 2c-d; $t(45) =$

-2.08, $p = .04$). This pattern suggests indeed an attention / choice effect rather than a value effect.

Review Figure 2 | Choice decoding analysis. Control analysis using only incorrectly answered trials. **a** Z-scores for the decoding probability difference for the on-screen high- vs. low-value stimuli based on decoding permutation test (see Methods). Occipital-temporal cortex represents the high-value stimulus significantly weaker than the low-value stimulus. **b** Visualization of the effect in a, showing the probabilities the decoder assigned to the stimuli (before the permutation test). **c** Z-scores for the decoding probability difference for the value-congruent off-screen high- vs. low-value stimuli based on decoding permutation test (see Methods). Occipital-temporal cortex represents the congruent high-value stimulus significantly weaker than the congruent low-value stimulus. **d** Visualization of the effect in c, showing the probabilities the decoder assigned to the stimuli (before the permutation test). * $p < .05$, *** $p < .001$

While we still think this is an interesting effect to report, we understand the reviewer's concern that the section might distract from the focus of the paper. We therefore decided to move this section, along with the new control analysis, to the Supplementary Material as Supplementary Fig. 8-10 (pages 54-57).

Methods, page 43:

Given that participants performed the task very well ($M = 87.70\%$), the stronger representation of the high-value option compared to the low-value option might be driven by selective attention towards the chosen option. To disentangle a value from an attention / choice effect, we repeated the analysis using only incorrectly answered trials (note the very low number of available incorrect trials). In this case, a stronger representation of the high vs. low value option would suggest a value effect while the opposite pattern, a stronger representation of the low-value option, would suggest an attention / choice effect.

1.4 Grid-like representation in vmPFC

I think it is quite surprising that there was no grid code in vmPFC especially given results from folks like the Rushworth lab, Behrens lab, and Boorman lab. I think it would be useful to devote at least some of this discussion to consideration of this inconsistency with the prior literature.

We thank the reviewer for raising this interesting point of discussion. Indeed, previous studies reported grid-like representations in vmPFC (e.g. Bao et al., 2019; Bongioanni et al., 2021; Constantinescu et al., 2016; Park et al., 2021) and we are also surprised that we did not observe such an effect in our study.

Interestingly, grid-like representations in vmPFC in previous studies have been detected using different analysis approaches. In our study, we estimated the putative entorhinal grid orientation using a set of training runs and then tested for a hexadirectional modulation in the remaining run **aligned to this entorhinal orientation at the whole-brain level**. While we did not observe any hexadirectional modulation of activity in vmPFC aligned to the entorhinal grid orientation, such an effect has been reported e.g. by Park et al. (2021). Another approach is to directly estimate the putative vmPFC grid orientation and test for a hexadirectional modulation **aligned to this vmPFC orientation**, for which effects have been reported e.g. by Constantinescu et al. (2016) and Bao et al. (2019). To investigate whether vmPFC activity in our study might be hexadirectionally modulated aligned to vmPFC's own estimated orientation, we defined two vmPFC ROIs and reran the hexadirectional analysis based on the estimated orientations of these ROIs. We defined the ROIs as spheres with a 7 mm radius 1) around the peak voxel of our value difference analysis in vmPFC (MNI peak voxel coordinates: 3,42,-8) and 2) around the peak voxel of the hexadirectional effect reported by Constantinescu et al. (2016) in vmPFC (MNI peak voxel coordinates: 16,54,-2). In both ROIs, there was no significant hexadirectional modulation aligned to the orientation of the respective ROI (Review Figure 3; vmPFC ROI based on value difference effect: $t(45) = -0.34$, $p = .63$; vmPFC ROI based on Constantinescu et al. (2016): $t(45) = -0.94$, $p = .82$). Hence, in both analysis approaches we found no evidence for a grid-like representation in vmPFC in our study.

Review Figure 3 | Grid-like representation in vmPFC. There was no significant grid-like hexadirectional modulation of activity aligned to the respective grid orientation of two vmPFC ROIs, the first one defined based on the value difference effect in vmPFC in our study (left) and the second one defined based on the hexadirectional effect in vmPFC reported by Constantinescu et al. (2016) (right).

We can only speculate as to why we found no evidence for a grid-like representation in vmPFC in our study. On the one hand, one could expect such an effect particularly in our study because vmPFC plays a prominent role in encoding value during decision making (Bartra et al., 2013; Boorman et al., 2009; De Martino et al., 2013; FitzGerald et al., 2009; Hunt et al., 2012; Knutson et al., 2005; Levy & Glimcher, 2012; O'Doherty et al., 2001; Padoa-Schioppa & Assad, 2006; Pelletier & Fellows, 2019; Plassmann et al., 2007) and we investigated an abstract value space based on the values of choice options. Alternatively, one could imagine that basic processing of value magnitudes per se in our task “overrides” more abstract neural representations of values. More specifically, we found a strong one-dimensional value difference signal in vmPFC and in line with the previous literature one could assume that this is one of vmPFC's predominant coding schemes. In light of this, it is conceivable that in our value-based decision making task with a high demand to track values, vmPFC engaged in its more prevalent coding scheme, which was encoding a one-dimensional value difference signal rather than the two-dimensional space. This idea is also in line with a recent study demonstrating a subjective value but no grid-like effect in vmPFC during a value-based intertemporal choice task (Lee et al., 2021). As suggested in the discussion of the manuscript, it is possible that different brain regions represent values in different, complementary ways, e.g., in a map-like format in the hippocampal-entorhinal system vs. a value difference or summary signal in vmPFC.

We included this new analysis on a possible grid-like representation in vmPFC in Supplementary Figure 4j and in the Methods section. We also extended the discussion regarding the lack of evidence for a grid-like representation in vmPFC in our study.

Discussion, pages 15-16:

In light of this, it is interesting to note that we found no evidence for a grid-like representation in vmPFC, where previous studies also reported such grid-like representations (Bao et al., 2019; Bongioanni et al., 2021; Constantinescu et al., 2016; Park et al., 2021). While keeping in mind that it is difficult to interpret the lack of evidence, we can only speculate as to why this discrepancy might have arisen. Based on a body of literature implicating vmPFC in representing the one-dimensional value difference between options during decision making (Bartra et al., 2013; Boorman et al., 2009; De Martino et al., 2013; FitzGerald et al., 2009; Hunt et al., 2012; Knutson et al., 2005; Lee et al., 2014; Levy & Glimcher, 2012; O'Doherty et al., 2001; Padoa-Schioppa & Assad, 2006; Pelletier & Fellows, 2019; Plassmann et al., 2007), one could assume that this is one of vmPFC's predominant coding schemes. It is conceivable that in our value-based decision making task with a high demand to track values, vmPFC engaged in its more prevalent coding scheme, which was encoding a one-dimensional value difference signal rather than the two-dimensional space. This idea is also in line with a recent study demonstrating a subjective value but no grid-like effect in vmPFC during a value-based intertemporal choice task (Lee et al., 2021). In this case, different value representations in the hippocampal-entorhinal system and vmPFC could serve complementary functions. For example, while the entorhinal value map could support the prediction of future values by facilitating computations of directions of and distances between value changes over time, other brain regions such as vmPFC might read out the resulting values, map them onto a single common scale for comparison and thus generate a one-dimensional signal of the value difference used for decision making.

Methods, page 34:

To further explore vmPFC representations, we defined two ROIs as spheres with a 7 mm radius 1) around the peak voxel of our value difference analysis in vmPFC (MNI peak voxel coordinates: 3,42,-8; 89 voxels) and 2) around the peak voxel of the hexadirectional effect reported by Constantinescu et al. (2016) in vmPFC (MNI peak voxel coordinates: 16,54,-2; 95 voxels).

Methods, page 39:

To further explore vmPFC representations, we repeated the cross-validated hexadirectional analysis while estimating the putative grid orientation in vmPFC (two vmPFC ROIs, see Region of interest (ROI) definition, analysis directly in MNI space). For each ROI, we then tested whether effect sizes were different from 0 using a one-sample t-test.

1.5 Color naming

Figure 1B: I'm not sure what the "arrow in petrol" means. Is "petrol" supposed to be a color? I've never heard of it. I think "light blue" would probably be a better descriptor.

We thank the reviewer for this comment. We changed the naming of the color to turquoise.

2 Reviewer 2

2.1 General remarks

The manuscript by Nitsch and colleagues reports a human fMRI study that measured neural representations of abstract value spaces during a decision making task. In the experiment, subjects were shown images corresponding to two different latent categories (A and B) along with values associated with each category. The critical feature of the task is that the values associated with these latent categories changed across trials in a consistent way, such that the changes could be described as a trajectory within the 2-d value space. The main question addressed in the study is whether this movement in an unseen 2-d value space was associated with grid-like responses (hexagonal coding) in the entorhinal cortex. While grid-like responses have been observed in entorhinal cortex for physical spaces in humans and rodents, and several fMRI studies have found grid-like coding for more conceptual spaces, the specific question of whether grid-like coding is observed for value spaces in human entorhinal cortex is novel. Indeed, the paper demonstrates grid-like effects within entorhinal cortex. These effects were selective to entorhinal cortex and also selective to a hexagonal coding scheme. Additional analyses show that value differences between categories (a 1 d representation) are reflected by widespread fMRI activations and that visual categories associated with higher rewards are preferentially represented in occipitotemporal cortex.

The paper is very well written, clearly organized, and well motivated from the prior literature. The key results are robust and there are a number of converging analyses to reinforce the main ideas. Thus, it is overall a very solid paper. The only notable weakness, in my view, is that the advance relative to prior studies is somewhat modest. There have now been several demonstrations of grid-like responses for non-physical spaces (in humans and rodents), so the extension to value-based decision making is not necessarily a major advance. Likewise, some of the secondary analyses are useful, but I think the framing of the results makes them sound a bit more interesting than they actually are. Thus, my assessment is that this is a very nice and rigorous paper that adds to an exciting literature (grid-like coding for conceptual information) but the degree of conceptual advance is perhaps borderline with respect to the standards for this journal.

We thank the reviewer for their positive evaluation of our manuscript and their thoughtful feedback. By addressing their comments, we have – amongst other aspects – included further control analyses to ensure the robustness of the value effects against confounds as well as control analyses for the choice decoding effects, which helped us restructure this section. We also highlight the novel contributions that we think this study adds to the literature in more detail. We believe that this has further strengthened our manuscript. Please find our detailed point-by-point responses below.

2.2 Advance to the study of grid-like representations

The main findings of grid-like coding in entorhinal cortex are, essentially, an extension of prior findings of similar effects for other conceptual spaces. I still find this striking and interesting, but the main goal of the current study appears to be to test whether this form of coding extends to value spaces. It is not clear that there is a meaningful alternative or that the paper tests competing ideas. Thus, it functions more of a nice demonstration / extension.

We thank the reviewer for their appreciation of our study and for raising this important point. While we agree that there are already several human studies showing grid-like representations for non-physical spaces (e.g. Bao et al., 2019; Constantinescu et al., 2016; Park et al., 2021; Viganò et al., 2021), many studies investigated spaces with feature dimensions that were still anchored to the physical world, e.g., the size of visual stimuli (Constantinescu et al., 2016). Even more abstract than stimulus features anchored to the physical world are values, i.e., rewards, associated with choice options. Furthermore, in the decision making literature, states in the world and values are usually considered different entities. However, it is conceivable that values constitute states themselves, which can be represented in a cognitive map. For these reasons, our human fMRI study, together with studies demonstrating grid- and place-like representations of values in macaques (Bongioanni et al., 2021; Knudsen & Wallis, 2021), provides important evidence for more abstract cognitive maps and contributes to merging ideas from two parallel research lines, namely value-based decision making and cognitive mapping in the hippocampal-entorhinal system.

To our knowledge, our study is the first human report of a grid-like value representation in the entorhinal cortex. The only other study on grid-like value representations from Bongioanni et al. (2021) found such an effect in macaques and in medial frontal cortex. Furthermore, while Bongioanni et al. (2021) conceptualized movement in a value space as the comparison of two static options (characterized by reward magnitude and probability), we investigated an even more abstract signal: the change in the relative value difference between options.

Lastly, we show in an exploratory analysis that – beyond simply encoding an abstract value space using a grid-like representation – the entorhinal grid system also adapts to informative properties of the space. More specifically, our results suggested an anchoring of grid orientations around 45° , which is a particularly informative reference direction through our value space. This is because it indicates that values of both options change at the same rate and – given that it is parallel to the 45° -diagonal of the value space – that there will be no switch of the more valuable option. While such an anchoring of grid orientations to an informative axis has recently been demonstrated for physical spaces in spatial navigation (Julian et al., 2018; Julian & Doeller, 2021; Navarro Schröder et al., 2020; Stensola et al., 2015), our results are indicative of the system's adaptive capacities even in more abstract spaces.

Taken together, our paper is therefore novel in three regards: we report (1) a more abstract grid-like value representation (2) in the entorhinal cortex which (3) aligns its orientation to the behaviorally most informative reference direction in the space. We believe that these findings provide important novel insights about abstract neural representations.

2.3 Choice decoding analysis

The stronger representation of higher value stimuli in occipitotemporal cortex makes sense, but is also not particularly exciting. This almost feels like a validation that subjects paid more attention to the stimuli that they were selecting. In other words, rather than framing this in terms of OTC 'caring' about value, it could simply be that OTC encoded what was attended/selected. Given that higher value items tended to be chosen much more often than lower value items, there is no simple way to disentangle value from choice. I do think the effects for value congruent stimuli (stimuli not actually on the screen) are neat, but again, I do not see how to tease apart a value representation, per se, from selective attention.

We thank the reviewer for bringing up this point (see also above for a similar point made by Reviewer 1, point 1.3). Given that participants performed the task very well ($M = 87.70\%$), we agree that the stronger representation of the high-value option compared to the low-value option might also reflect selective attention towards the chosen option.

In our view, the only possibility to disentangle a value from an attention / choice effect is to conduct the decoding analysis using only incorrectly answered trials. In this case, a stronger representation of the high vs. low value option would suggest a value effect while the opposite pattern, a stronger representation of the low-value option, would suggest an attention / choice effect. We conducted this control analysis and despite the very low number of available incorrect trials, we observed significantly lower probabilities for the high-value vs. the low-value stimulus, both when comparing on-screen stimuli (Review Figure 4a-b; $t(45) = -5.57$, $p < .001$) as well as when comparing congruent off-screen stimuli (Review Figure 4c-d; $t(45) = -2.08$, $p = .04$). This pattern suggests indeed an attention / choice effect rather than a value effect.

Review Figure 4 | Choice decoding analysis. Control analysis using only incorrectly answered trials. **a** Z-scores for the decoding probability difference for the on-screen high- vs. low-value stimuli based on decoding permutation test (see Methods). Occipital-temporal cortex represents the high-value stimulus significantly weaker than the low-value stimulus. **b** Visualization of the effect in a, showing the probabilities the decoder assigned to the stimuli (before the permutation test). **c** Z-scores for the decoding probability difference for the value-congruent off-screen high- vs. low-value stimuli based on decoding permutation test (see Methods). Occipital-temporal cortex represents the congruent high-value stimulus significantly weaker than the congruent low-value stimulus. **d** Visualization of the effect in c, showing the probabilities the decoder assigned to the stimuli (before the permutation test). * $p < .05$, *** $p < .001$

While we still think this is an interesting effect to report, we understand the concern raised by Reviewer 1 that the section might distract from the focus of the paper. We therefore decided to move this section, along with the new control analysis, to the Supplementary Material as Supplementary Fig. 8-10 (pages 54-57).

Methods, page 43:

Given that participants performed the task very well ($M = 87.70\%$), the stronger representation of the high-value option compared to the low-value option might be driven by selective attention towards the chosen option. To disentangle a value from an attention / choice effect, we repeated the analysis using only incorrectly answered trials (note the very low number of available incorrect trials). In this case, a stronger representation of the high vs. low value option would suggest a value effect while the opposite pattern, a stronger representation of the low-value option, would suggest an attention / choice effect.

2.4 Controls for the prospective value difference effects

Likewise, the prospective value effects (1d space) are worth reporting, but not particularly exciting because it is very hard to disentangle a representation of value, per se, from the decision difficulty. I do appreciate that the authors controlled for RT, but it is hard to fully control for difficulty (e.g., the authors used log-transformed RT, which

makes an assumption about how difficulty scales with RT). The fact that the 1d value effects were so widespread in the brain also suggests that something like effort or difficulty is a reasonable interpretation. It is not clear that there is any way, in the current design, to completely tease apart the value estimate from other things that are bound to be confounded with value estimates.

We thank the reviewer for this suggestion. In addition to controlling for reaction time in the value analyses, we thought of two more approaches to control for potential confounds such as difficulty or effort.

First, we investigated the value effects using only correct trials, with the goal to eliminate confounds associated specifically with incorrect trials such as difficulty. We thus reran both value analyses, the first one assessing the value difference between options during choices and the second one assessing particularly the prospective component of the value difference, by restricting the parametrically modulated value regressors only to correctly answered choice time points. We observed similar effects as in the original analyses for both the value difference as well as the prospective component (Review Figure 4).

Secondly, we thought about including the distance of the choice location from the 45°-diagonal as another regressor to control for difficulty. In the case of the actual value difference between options during choices, this will not work because – given our design of a value space – the distance between the choice location and the diagonal reflects inherently a measure of the unsigned value difference. Indeed, when we built a GLM including the value and distance regressors, the variance inflation factors of these regressors were very high ($M = 127.33$; with $VIF > 4-5$ indicating multicollinearity), suggesting multicollinearity. However, when investigating particularly the prospective component of the value difference, i.e., the influence of values estimated by the prospective Rescorla-Wagner model over values estimated by the original (non-prospective) Rescorla-Wagner model, there should be no relationship with the distance between the choice location and the diagonal. Indeed, when we built a GLM including the prospective value and distance regressors, the variance inflation factors of these regressors indicated no problem with multicollinearity ($M = 1.65$). We thus conducted this control analysis and observed similar effects for the prospective component of the value difference as in the original analysis (Review Figure 5).

Review Figure 4 | Controls for the prospective value difference effects: including only correct trials. A Control analysis: Modulation of activity by the difference between model-derived chosen vs. unchosen value during choices when including only correct trials. Clusters depicted survive whole-brain correction ($p_{FWE} < .05$, TFCE). Statistical image is displayed on the MNI template. **b** Control analysis: Modulation of activity by the prospective component of the value difference during choices when including only correct trials. The prospective component refers to the influence of values estimated by the prospective Rescorla-Wagner model over values estimated by the original (non-prospective) Rescorla-Wagner model. Clusters depicted survive whole-brain correction ($p_{FWE} < .05$, TFCE). Statistical image is displayed on the MNI template.

Review Figure 5 | Controls for the prospective value difference effects: controlling for the distance between the choice location and the 45°-diagonal. Control analysis: Modulation of activity by the prospective component of the value difference during choices after controlling for the distance between the choice location and the 45°-diagonal. The prospective component refers to the influence of values estimated by the prospective Rescorla-Wagner model over values estimated by the original (non-prospective) Rescorla-Wagner model. Clusters depicted survive whole-brain correction ($p_{FWE} < .05$, TFCE). Statistical image is displayed on the MNI template.

Taken together, we observed that our value effects were robust against potential confounds using three different analysis approaches – controlling for reaction time, correct / incorrect trials and the distance between the choice location and the 45°-diagonal. We are therefore

confident that these activity patterns reflect a value difference signal. We now include these new controls in the Results and Methods sections and as Supplementary Figures 6-7.

Results, pages 12-13:

These effects were still present when controlling for reaction time (Supplementary Fig. 5a) and when restricting the analysis to correct trials only (Supplementary Fig. 6a).

[...]

This prefrontal cluster was still present when controlling for reaction time (Supplementary Fig. 5b), restricting the analysis to correct trials only (Supplementary Fig. 6b) and controlling for the distance between the choice location and the 45°-diagonal (Supplementary Fig. 7).

Methods, pages 39-40:

In three separate control analyses, 1) we added an additional parametrically modulated regressor for choice time points reflecting reaction time, 2) we restricted the parametrically modulated value regressors to correct trials only and 3) we added an additional parametrically modulated regressor for choice time points reflecting the distance between the choice location and the 45°-diagonal. Reaction times were log-transformed and demeaned, the distance between the choice location and the diagonal was demeaned.

2.5 Strategies to solve the task

I am wondering if subjects tended to settle into a strategy of explicitly “doing the math” to make their decisions. For example, after the 2nd time point, subjects know the increment and direction of change (e.g., +5 for variable A and -9 for variable B). Did subjects report tracking this explicitly? At least to me, this seems like it would be the best way to perform the task. Obviously, this strategy would not involve an integration into a 2d space. As such, I don’t see how this strategy would explain the fMRI results. So, my question is not necessarily a concern; rather, I am just curious what strategy subjects tended to use and whether this might constrain interpretations at all? The Discussion says that subjects did not report using a 2D (integrated) space, but I did not see any mention of strategies that subjects DID report using.

We thank the reviewer for this question. We did ask participants at the end of the study which strategies they used to solve the task. Then we told them about the underlying two-dimensional value space and asked whether they imagined such a space. Unfortunately, the first part of this debriefing was only verbal so that we cannot exactly quantify the strategies. Many participants reported imagining separate number lines and some participants mentioned trying to calculate the value changes. Some participants also reported that they felt that the value and stimulus changes happened very fast (each time point was presented for 2.5 s). In the second part of the debriefing, we recorded answers to the question whether they imagined the underlying two-dimensional value space to potentially exclude those participants from the analysis. No participant reported doing so. In summary, it is striking that the neural data suggests a two-dimensional spatial representation even though participants did not report

using a space/map-based strategy. Examining the relation between explicitly accessible strategies and neural computations as reported here is an interesting topic for future research and underscores the need to investigate specifically humans as opposed to animals, including non-human primates.

We now describe participants' reports during the debriefing in more detail in the revised paragraph in the Methods section.

Methods, page 22:

At the end of the study, participants were asked which strategies they used to solve the task. Then they were told about the underlying two-dimensional value space and asked whether they imagined such a space. The first question about the strategies was only verbal so that we cannot exactly quantify them. While participants mentioned, amongst others, imagining separate number lines and trying to calculate the value changes, they also reported that they felt that the value and stimulus changes happened very fast. For the second question – whether participants imagined the underlying two-dimensional value space – we recorded answers to potentially exclude those participants from the analysis. No participant reported having imagined the underlying two-dimensional value space.

2.6 Typos

p. 8: "We found that the likelihood of correct choices increased with increasing distance of the choice location to the diagonal (Fig. 2c-d; $t(44) = 8.03, p < .001$)." Should this instead say "distance of the choice location FROM the diagonal"? As written, it sounds like accuracy increased as choices got closer to the diagonal, but I assume the authors mean the opposite?

Fig. 3 legend: panel "f" is referred to as panel "e"

We thank the reviewer for catching the typos. We changed all instances of the expression "distance to the diagonal" to "distance from the diagonal". We also changed the Figure 3 legend to refer to panel f.

3 References

- Bao, X., Gjorgieva, E., Shanahan, L. K., Howard, J. D., Kahnt, T., & Gottfried, J. A. (2019). Grid-like Neural Representations Support Olfactory Navigation of a Two-Dimensional Odor Space. *Neuron*, 102(5), 1066-1075.e5. <https://doi.org/10.1016/j.neuron.2019.03.034>
- Bartra, O., McGuire, J. T., & Kable, J. W. (2013). The valuation system: A coordinate-based meta-analysis of BOLD fMRI experiments examining neural correlates of subjective value. *NeuroImage*, 76, 412–427. <https://doi.org/10.1016/j.neuroimage.2013.02.063>
- Bongioanni, A., Folloni, D., Verhagen, L., Sallet, J., Klein-Flügge, M. C., & Rushworth, M. F. S. (2021). Activation and disruption of a neural mechanism for novel choice in monkeys. *Nature*, 591(7849), Article 7849. <https://doi.org/10.1038/s41586-020-03115-5>
- Boorman, E. D., Behrens, T. E. J., Woolrich, M. W., & Rushworth, M. F. S. (2009). How Green Is the Grass on the Other Side? Frontopolar Cortex and the Evidence in Favor of Alternative Courses of Action. *Neuron*, 62(5), 733–743. <https://doi.org/10.1016/j.neuron.2009.05.014>
- Constantinescu, A. O., O'Reilly, J. X., & Behrens, T. E. J. (2016). Organizing conceptual knowledge in humans with a gridlike code. *Science*, 352(6292), 1464–1468. <https://doi.org/10.1126/science.aaf0941>
- De Martino, B., Fleming, S. M., Garrett, N., & Dolan, R. J. (2013). Confidence in value-based choice. *Nature Neuroscience*, 16(1), Article 1. <https://doi.org/10.1038/nn.3279>
- FitzGerald, T. H. B., Seymour, B., & Dolan, R. J. (2009). The Role of Human Orbitofrontal Cortex in Value Comparison for Incommensurable Objects. *Journal of Neuroscience*, 29(26), 8388–8395. <https://doi.org/10.1523/JNEUROSCI.0717-09.2009>

- Hunt, L. T., Kolling, N., Soltani, A., Woolrich, M. W., Rushworth, M. F. S., & Behrens, T. E. J. (2012). Mechanisms underlying cortical activity during value-guided choice. *Nature Neuroscience*, *15*(3), Article 3. <https://doi.org/10.1038/nn.3017>
- Julian, J. B., & Doeller, C. F. (2021). Remapping and realignment in the human hippocampal formation predict context-dependent spatial behavior. *Nature Neuroscience*, *24*(6), 863–872. <https://doi.org/10.1038/s41593-021-00835-3>
- Julian, J. B., Keinath, A. T., Frazzetta, G., & Epstein, R. A. (2018). Human entorhinal cortex represents visual space using a boundary-anchored grid. *Nature Neuroscience*, *21*(2), Article 2. <https://doi.org/10.1038/s41593-017-0049-1>
- Knudsen, E. B., & Wallis, J. D. (2021). Hippocampal neurons construct a map of an abstract value space. *Cell*, *184*(18), 4640–4650.e10. <https://doi.org/10.1016/j.cell.2021.07.010>
- Knutson, B., Taylor, J., Kaufman, M., Peterson, R., & Glover, G. (2005). Distributed Neural Representation of Expected Value. *Journal of Neuroscience*, *25*(19), 4806–4812. <https://doi.org/10.1523/JNEUROSCI.0642-05.2005>
- Lee, S. W., Shimojo, S., & O'Doherty, J. P. (2014). Neural Computations Underlying Arbitration between Model-Based and Model-free Learning. *Neuron*, *81*(3), 687–699. <https://doi.org/10.1016/j.neuron.2013.11.028>
- Lee, S., Yu, L. Q., Lerman, C., & Kable, J. W. (2021). Subjective value, not a gridlike code, describes neural activity in ventromedial prefrontal cortex during value-based decision-making. *NeuroImage*, *237*, 118159. <https://doi.org/10.1016/j.neuroimage.2021.118159>
- Levy, D. J., & Glimcher, P. W. (2012). The root of all value: A neural common currency for choice. *Current Opinion in Neurobiology*, *22*(6), 1027–1038. <https://doi.org/10.1016/j.conb.2012.06.001>

- Navarro Schröder, T. N., Towse, B. W., Nau, M., Burgess, N., Barry, C., & Doeller, C. F. (2020). Environmental anchoring of grid-like representations minimizes spatial uncertainty during navigation. *BioRxiv*, 166306. <https://doi.org/10.1101/166306>
- O'Doherty, J., Kringelbach, M. L., Rolls, E. T., Hornak, J., & Andrews, C. (2001). Abstract reward and punishment representations in the human orbitofrontal cortex. *Nature Neuroscience*, 4(1), Article 1. <https://doi.org/10.1038/82959>
- Padoa-Schioppa, C., & Assad, J. A. (2006). Neurons in the orbitofrontal cortex encode economic value. *Nature*, 441(7090), Article 7090. <https://doi.org/10.1038/nature04676>
- Park, S. A., Miller, D. S., & Boorman, E. D. (2021). Inferences on a multidimensional social hierarchy use a grid-like code. *Nature Neuroscience*, 24(9), 1292–1301. <https://doi.org/10.1038/s41593-021-00916-3>
- Pelletier, G., & Fellows, L. K. (2019). A Critical Role for Human Ventromedial Frontal Lobe in Value Comparison of Complex Objects Based on Attribute Configuration. *The Journal of Neuroscience*, 39(21), 4124–4132. <https://doi.org/10.1523/JNEUROSCI.2969-18.2019>
- Plassmann, H., O'Doherty, J., & Rangel, A. (2007). Orbitofrontal Cortex Encodes Willingness to Pay in Everyday Economic Transactions. *Journal of Neuroscience*, 27(37), 9984–9988. <https://doi.org/10.1523/JNEUROSCI.2131-07.2007>
- Stensola, T., Stensola, H., Moser, M.-B., & Moser, E. I. (2015). Shearing-induced asymmetry in entorhinal grid cells. *Nature*, 518(7538), Article 7538. <https://doi.org/10.1038/nature14151>
- Viganò, S., Rubino, V., Soccio, A. D., Buiatti, M., & Piazza, M. (2021). Grid-like and distance codes for representing word meaning in the human brain. *NeuroImage*, 232, 117876. <https://doi.org/10.1016/j.neuroimage.2021.117876>

REVIEWERS' COMMENTS

Reviewer #1 (Remarks to the Author):

The authors have addressed all my prior concerns

Reviewer #2 (Remarks to the Author):

The authors have addressed my concerns (which were only minor). I also reviewed the concerns raised by the other reviewer and feel that those have been addressed as well. This is a very nice paper that will be a significant contribution. I have no remaining concerns.